# Improving *day-ahead* Solar Irradiance Time Series Forecasting by Leveraging Spatio-Temporal Context

**Oussama Boussif** [1,2] *       **Ghait Boukachab** [1,3] *       **Dan Assouline** [1,2] *
**Stefano Massaroli**[1,2]     **Tianle Yuan**[4]     **Loubna Benabbou**[3]     **Yoshua Bengio**[1,2]

[1]Mila - Québec AI Institute         [3]Université du Québec à Rimouski
[2]Université de Montréal         [4]NASA Goddard Space Flight Center

## Abstract

Solar power harbors immense potential in mitigating climate change by substantially reducing $CO_2$ emissions. Nonetheless, the inherent variability of solar irradiance poses a significant challenge for seamlessly integrating solar power into the electrical grid. While the majority of prior research has centered on employing purely time series-based methodologies for solar forecasting, only a limited number of studies have taken into account factors such as cloud cover or the surrounding physical context. In this paper, we put forth a deep learning architecture designed to harness spatio-temporal context using satellite data, to attain highly accurate *day-ahead* time-series forecasting for any given station, with a particular emphasis on forecasting Global Horizontal Irradiance (GHI). We also suggest a methodology to extract a distribution for each time step prediction, which can serve as a very valuable measure of uncertainty attached to the forecast. When evaluating models, we propose a testing scheme in which we separate particularly difficult examples from easy ones, in order to capture the model performances in crucial situations, which in the case of this study are the days suffering from varying cloudy conditions. Furthermore, we present a new multi-modal dataset gathering satellite imagery over a large zone and time series for solar irradiance and other related physical variables from multiple geographically diverse solar stations. Our approach exhibits robust performance in solar irradiance forecasting, including zero-shot generalization tests at unobserved solar stations, and holds great promise in promoting the effective integration of solar power into the grid.

## 1 Introduction

Solar power has become an increasingly important source of renewable energy in recent years, with the potential to help mitigate the effects of climate change by reducing greenhouse gas emissions (Doblas-Reyes et al., 2021; IEA, 2021). However, the variability of solar irradiance - the amount of solar radiation that reaches the earth's surface - presents a challenge for integrating solar power into the grid. Accurate forecasting of solar irradiance can assist grid operators in dealing with the variability of solar power, leading to a more efficient and dependable integration of solar power into the grid. As a result, this can help to reduce the requirement for costly and environmentally damaging backup power sources.

---

*Equal contribution.

Corresponding authors: oussama.boussif@mila.quebec, ghait.boukachab@mila.quebec, dan.assouline@mila.quebec

Code & dataset: `https://github.com/gitbooo/CrossViVit`

37th Conference on Neural Information Processing Systems (NeurIPS 2023).

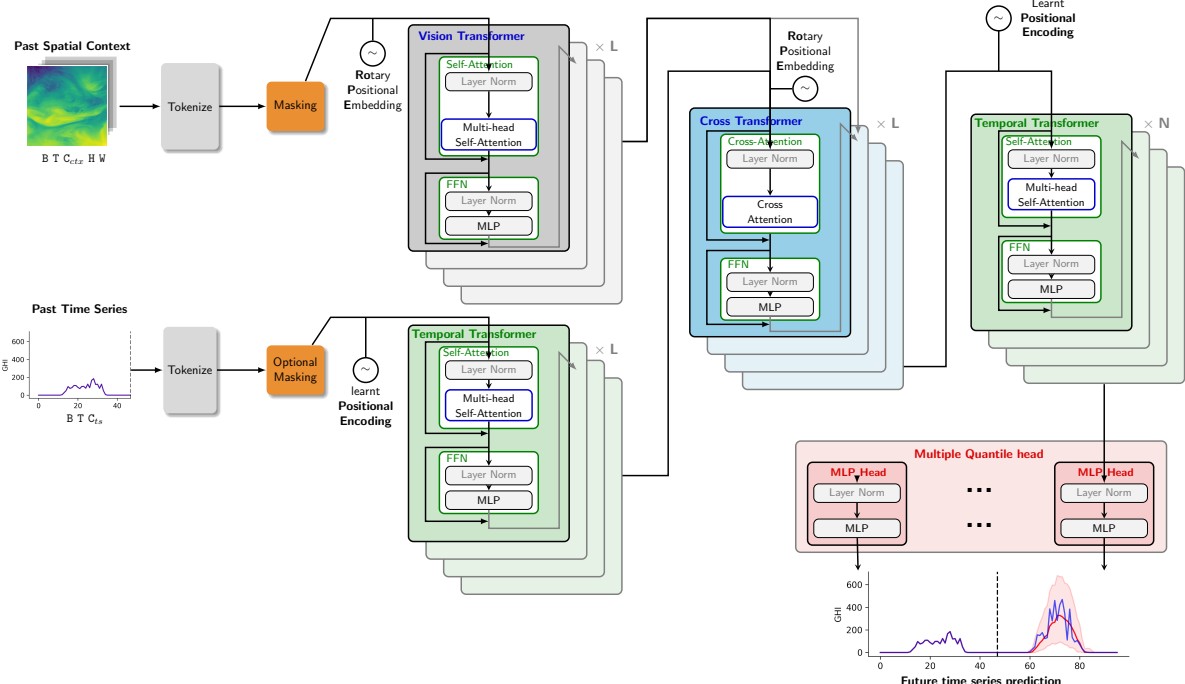

Figure 1: **CrossViViT architecture, in its Multi-Quantile version.** (1) The spatio-temporal context videos are tokenized, partially masked, and encoded with a vision transformer, using Rotary Positional Embedding, (2) The time series are tokenized and encoded in parallel with a transformer, (3) Both resulting latents are fed into $L$ layers of a *cross transformer* to mix them into one latent, (4) the output is passed into another transformer decoder, (5) and passed to multiple MLP heads which predict multiple quantiles, forming a prediction interval for each day-ahead time-series prediction.

Solar irradiance is influenced by a range of factors, including the time of day, the season, weather patterns, and the position of the sun in the sky. However, one of the most significant factors affecting solar irradiance variability is cloud cover. Clouds can block or scatter solar radiation, leading to rapid changes in solar irradiance at the earth's surface. Forecasting solar irradiance accurately thus requires modeling cloud cover, as well as accounting for the inherent variability of the system.

While a lot of previous work has focused on using pure time-series approaches to forecast solar irradiance (Yang et al., 2022), few have incorporated cloud cover (Nielsen et al., 2021; Bone et al., 2018; Si et al., 2021) and even fewer tackled the challenging **day-ahead forecasting**, often focusing on the easier very short term predictions (2 to 4 hours). When forecasting solar irradiance at a specific station, relying solely on its local physical variables is insufficient due to the significant spatial variability of cloud cover. To accurately anticipate the impact of clouds on incoming solar radiation, it is necessary to consider their motion and trajectory within a larger spatial context encompassing the station. In this paper, we incorporate satellite imaging to forecast solar irradiance at any chosen station and propose a multi-modal architecture that can in principle be used to **forecast any physical variable**. This study reveals the inadequacy of conventional testing schemes using conventional metrics like MAE or RMSE on an entire dataset for evaluating solar irradiance forecasting models. They fail to capture the models' performance in crucial cloud-related scenarios. Hence, we emphasize the necessity of a separate evaluation approach to address the complexity of cloud impact on solar irradiance variability. To allow for the metrics to show such performances by limiting the smoothing effect of the averaging, we propose a testing scheme based on multiple splits of the test data, separating particularly difficult examples from easy ones, for the task at hand. This is particularly important for practical downstream tasks related to solar irradiance estimation, such as the monitoring of solar power plants, where the correct prediction of such difficult examples can have a very high impact, as easy examples (in the presence of light cloud cover) can be treated fairly well by very simple models (such as a clear-sky or persistence model, as presented in section 5.1).

**Contributions:** Our focus is on the application of machine learning to time-series forecasting, with a particular emphasis on utilizing multi-modal spatio-temporal data that includes physical, weather, and remotely sensed variables. The main contributions of this paper can be summarized as follows:

- We present a deep learning architecture called CrossViViT, designed to leverage spatio-temporal context (such as satellite data) in order to achieve highly accurate medium-term (1 day horizon) time-series forecasting at any given station. This paper focuses specifically on forecasting Global Horizontal Irradiance (GHI).

- We present a Multi-Quantile version of the model which allows to extract uncertainty estimation attached to each prediction. This methodology, although applied here in our context, should be applicable to any forecasting task and framework.

- We present a multi-modal dataset that combines satellite imagery with solar irradiance and other physical variables. The dataset covers a period of 14 years, from 2008 to 2022, and includes data from six geographically diverse solar stations. This dataset is unique in its combination of diverse variables and long time span, and is intended to facilitate the development and evaluation of new multi-modal forecasting models for solar irradiance.

- We propose a forecasting testing scheme based on multiple time splits of the test data, separating particularly difficult examples from easy ones, therefore allowing to capture the models' performances in problematic examples.

- We experimentally show that the proposed approach can generalize to a new station not seen during training in a zero-shot generalization forecasting setting.

## 2   Related works

**Machine Learning for time-series forecasting** Deep learning approaches have gained popularity for time-series forecasting in recent years due to their ability to model complex nonlinear relationships and capture temporal dependencies. These approaches have demonstrated superior performance compared to traditional statistical methods, motivating further research in this area. In a recent survey (Wen et al., 2022), it was found that transformers, renowned for their success in natural language processing and computer vision, were also effective for time-series analysis. The authors discussed the strengths and limitations of transformers and compared the structure and performance of recent transformer-based architectures on a benchmark weather dataset (Zhou et al., 2021). The particular case of solar irradiance forecasting represents an interesting application for time-series models (Wang et al., 2019a; Narvaez et al., 2021; Alzahrani et al., 2017). One recent study developed a multi-step attention-based model for solar irradiance forecasting that generates deterministic predictions and quantile predictions as well (Sharda et al., 2021). In a similar perspective, Jønler et al. (2023) developed a probabilistic solar irradiance transformer that incorporates gated recurrent units and temporal convolution networks, demonstrating strong performance for short-term horizons.

**Context mixing / Multimodal learning for time-series forecasting** Previous studies highlight the potential of time-series methods for solar irradiance forecasting, emphasizing the significance of short-term horizons in solar energy management. However, day-ahead forecasting remains challenging due to the influence of cloud cover on surface irradiance (Bone et al., 2018; Si et al., 2021), a problem which we aim to address in this paper. Thus, it is crucial to account for cloud effects in solar irradiance forecasting regardless of the chosen method. For instance, Zhang et al. (2023) investigated the impact of cloud movement on irradiance prediction and proposed an approach to automatically learn the relationship between sky image appearance and solar irradiance. A concurrent work (Liu et al., 2023) proposed a multimodal-learning framework for ultra-short-term (10min-ahead) solar irradiance forecasting. They used Informer (Zhou et al., 2021) to encode historical time-series data, then utilized Vision Transformer (Dosovitskiy et al., 2020) to handle sky images. Finally, they employed cross-attention to couple the two modalities. The studies discussed above highlight the potential of incorporating external data sources, such as sky images and satellite images, in combination with time-series approaches to improve the accuracy of solar forecasting.

**Operator Learning** Utilizing available satellite imagery to forecast GHI over a region presents limitations as it may not capture clouds that exist at a resolution beyond that of the satellite data. To ensure accurate forecasting of quantities of interest, the ability to query the model at any possible resolution and any point within the domain becomes crucial. Recent advancements have witnessed

the rise of algorithms focusing on learning operators capable of mapping across functional spaces, with a focus on solving partial differential equations (PDE) (Lu et al., 2019; Li et al., 2021; Kovachki et al., 2021; Li et al., 2020). These operators can effectively map initial conditions to PDE solutions, making it possible to query the learned solution theoretically anywhere within its domain. Fourier Layers, developed by Li et al. (2021), enable zero-shot prediction on both uniform and non-uniform grids with learnable deformations (Li et al., 2022). Pathak et al. (2022) replace attention in ViT (Dosovitskiy et al., 2020) with Fourier layer mixing for competitive weather forecasting results with faster inference. MeshFreeflowNet (Jiang et al., 2020) learns high-resolution frames from corresponding lower resolution ones by querying the model at any point of the domain for irregular grids. Similarly, Boussif et al. (2022) employ message passing with a low-resolution graph for zero-shot super-resolution PDE learning. Additionally, message passing neural PDE solvers (Brandstetter et al., 2022) exhibit spatio-temporal multi-scale capabilities benefiting from long-expressive memory (Equer et al., 2023; Rusch et al., 2022). We note that while these approaches were developed for PDEs in mind, they can still be used for weather-related applications.

**Uncertainty estimation** When performing solar irradiance forecasting, deterministic forecasts are not sufficient to characterize the inherent variance and uncertainty in solar irradiance data. Probabilistic forecasts, providing uncertainty information, are crucial for energy system management (Wang et al., 2019b). (Doubleday et al., 2020) and (Yagli et al., 2020) benchmark solar forecasting methods, emphasizing calibration and sharpness in prediction intervals. Specifically in short term solar irradiance forecasting, Zelikman et al. (2020) delves into post-hoc calibration for better predictions. Turkoglu et al. (2022) introduces FiLM-Ensemble, balancing predictive accuracy and calibration in uncertainty estimation. The deep ensembles approach by Lakshminarayanan et al. (2016) aggregates neural network predictions, capturing both data noise and model uncertainties. Few studies tackle uncertainty evaluation in a regression setting, rather than in classification one, which consists in the estimation of prediction intervals. Sønderby et al. (2020), performing precipitation forecasting, suggests an easy solution: the output is separated in 512 bins and the model predicts the precipitation rate (probability) in each of the bins, resulting ultimately in a distribution. Other methodologies include bootstrapping and ensembling methods, drawing a distribution out of multiple predictions from submodels, following the spirit of Quantile Regression Forests (Meinshausen and Ridgeway, 2006). The evolving landscape of solar irradiance forecasting underscores the importance of a calibrated, comprehensive, and robust probabilistic approach to address inherent uncertainties.

## 3 Methodology

We develop a framework for solar irradiance time-series forecasting, incorporating spatio-temporal context alongside historical time-series data from multiple stations. This framework is inspired by recent advancements in video transformer models (Arnab et al., 2021; Feichtenhofer et al., 2022) and multi-modal models that leverage diverse data sources such as images and time series (Liu et al., 2023). To establish the foundation for our framework, we provide a brief overview of the Rotary Positional Embedding (Su et al., 2021). Subsequently, we present the proposed architecture, CrossViViT, in detail, outlining its key components and design principles. Details about the ViT (Dosovitskiy et al., 2020) and ViViT architectures can be found in the Appendix.

### 3.1 Rotary Positional Embedding

As we are dealing with permutation-variant data, assigning positions to patches before using attention is necessary. A common approach is to use additive positional embedding, which considers the absolute positions of the patches added into the input sequence. However, for our case, it is more appropriate to have dot-product attention depend on the relative "distance" between the patches. This is because the station should only be concerned with the distance and direction of nearby clouds, and therefore, a relative positional embedding is more sensible.

To this end we make use of RoPE (Su et al., 2021) to mix the station time series and the context. The station and each patch in the context are assigned their *normalized* latitude and longitude coordinates in $[-1, 1]$ which are used as positions for RoPE. Details about the formulation used are left to the Appendix.

### 3.2 Cross Video Vision Transformer for time-series forecasting (CrossViViT)

**CrossViViT.** Our approach aims to integrate the available historical time-series data with video clips of spatio-temporal physical context to enhance the prediction of future time series for solar irradiance. The overall methodology, depicted in Figure 1, can be summarized as follows:

1. **Tokenizing**: The video context $\mathbf{V} \in \mathbb{R}^{T \times C_{ctx} \times H \times W}$, with $T$ frames for each of the $C_{ctx}$ channels, and $H$ and $W$ respectively the height and width of the video images, is divided into $N_p$ non-overlapping patches and linearly projected into a sequence of $d$-dimensional context tokens $\mathbf{z}^{ctx} \in \mathbb{R}^{T \times N_p \times d}$. We use the *Uniform frame sampling* ViViT scheme Arnab et al. (2021) to embed the videos we have at hand, the frames being concatenated along the batch dimension, and the sample frequency being defined at 30 minutes. The historical time series $\mathbf{t} \in \mathbb{R}^{T \times C_{ts}}$ are linearly projected into a sequence of $d$-dimensional time-series tokens $\mathbf{z}^{ts} \in \mathbb{R}^{T \times d}$. We augment the context tokens with RoPE, as presented in section 3.1, and a learnt positional encoding for the time-series tokens.

2. **Masking**: As a regularizing mechanism, we allow the model to mask a portion of the past time series and the video context before adding the positional encodings. During the training phase, a masking ratio $m_{ctx}$ is randomly sampled from a uniform distribution $U(0, 0.99)$ for the context, and the corresponding patches are masked accordingly. We also explored masking the time series to encourage the model to rely more on the context but in practice, no masking gave the best performance. We note that during inference, no masking is applied.

3. **Encoding:** We encode the time series and the past video context separately with two transformer architectures: a spatio-temporal encoder similar to a ViT for the video context, and a multi layer transformer for the input time series. More specifically, the context tokens alone and time series alone are passed through $L$ separate transformer layers (we keep the same number of layers $L$ for both encoders), including Multi-Head Self-Attention (MSA), LayerNorm (LN) and Multi-Layer Perceptron (MLP) blocks, so that for each layer $l$, we perform the following operations:

$$\mathbf{y}_l^{ctx} = \text{MSA}(\text{LN}(\mathbf{z}_l^{ctx})) + \mathbf{z}_l^{ctx} \qquad\qquad \mathbf{y}_l^{ts} = \text{MSA}(\text{LN}(\mathbf{z}_l^{ts})) + \mathbf{z}_l^{ts} \qquad (1)$$

$$\mathbf{z}_{l+1}^{ctx} = \text{MLP}(\text{LN}(\mathbf{y}_l^{ctx})) + \mathbf{y}_l^{ctx} \qquad\qquad \mathbf{z}_{l+1}^{ts} = \text{MLP}(\text{LN}(\mathbf{y}_l^{ts})) + \mathbf{y}_l^{ts} \qquad (2)$$

4. **Mixing:** We combine the resulting context and time-series latents, respectively $\mathbf{z}_L^{ctx}$ and $\mathbf{z}_L^{ts}$, within $L$ layers of a Transformer with Cross Attention (CA) Vaswani et al. (2017) (we keep the same number of layers in the entire encoder-mixer architecture). After adding ROPE, the two $L$-th layers are mixed with CA and passed through an MLP block. The output of each layer becomes a mixed latent which is in turn mixed with the context latent $\mathbf{z}_L^{ctx}$ and again passed through a block of MLP. Formally, the following operations are performed respectively at the first layer (left equations) and on the remaining layers (right equations) of the CA:

$$\mathbf{y}_1^{mix} = \text{CA}(\text{LN}(\mathbf{z}_L^{ctx}, \mathbf{z}_L^{ts})) + \mathbf{z}_L^{ts} \qquad\qquad \mathbf{y}_l^{mix} = \text{CA}(\text{LN}(\mathbf{z}_L^{ctx}, \mathbf{z}_l^{mix})) + \mathbf{z}_l^{mix} \qquad (3)$$

$$\mathbf{z}_2^{mix} = \text{MLP}(\text{LN}(\mathbf{y}_1^{mix})) + \mathbf{y}_1^{mix} \qquad\qquad \mathbf{z}_{l+1}^{mix} = \text{MLP}(\text{LN}(\mathbf{y}_l^{mix})) + \mathbf{y}_l^{mix} \qquad (4)$$

5. **Decoding:** The sequence of mixed tokens $\mathbf{z}_L^{mix}$ returned by the layers of Cross Transformer is then passed through $N$ layers of another Transformer as a decoder, before adding a learnt positional embedding to the token sequence. Each layer $n$ of the Transformer is again formed by MSA, LN and MLP blocks:

$$\mathbf{y}_n = \text{MSA}(\text{LN}(\mathbf{z}_n)) + \mathbf{z}_n \qquad (5)$$

$$\mathbf{z}_{n+1} = \text{LN}(\text{MLP}(\text{LN}(\mathbf{y}_n)) + \mathbf{y}_n) \qquad (6)$$

The output decoded sequence $\mathbf{z}_N$ is passed through a final MLP head to output the final predicted future time series $\mathbf{t}_{pred} \in \mathbb{R}^{T \times C_{ts}}$.

### 3.3 Multi-Quantiles: Extracting prediction intervals

To obtain prediction intervals for each forecasted value, we propose a easy methodology which can be used for any deep learning forecasting model, here resulting in an alternative version of the

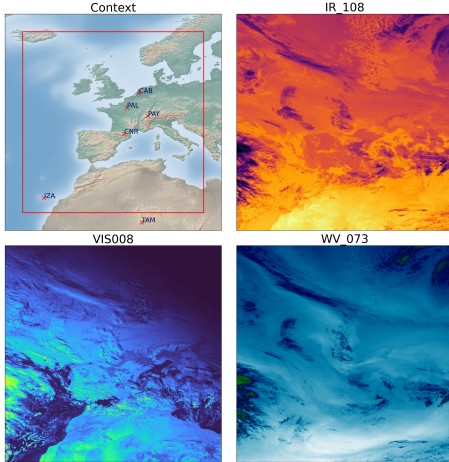

| Station | Latitude | Longitude |
|---|---|---|
| Cabauw (CAB) (Knap, 2007) | $51°58'N$ | $4°55'E$ |
| Cener (CNR) (Olano, 2022) | $42°48'N$ | $1°36'W$ |
| Izana (IZA) (Cuevas-Agulló, 2014) | $28°18'N$ | $16°29'W$ |
| Palaiseau (PAL) (Haeffelin, 2014) | $48°42'N$ | $2°12'E$ |
| Payerne (PAY) (Vuilleumier, 2018) | $46°48'N$ | $6°56'E$ |
| Tamanrasset (TAM) (Baika, 2023) | $22°47'N$ | $5°31'E$ |

Figure 2: **Stations and satellite data.** *Left*: Location of the six meteorological stations considered in the study with the red border indicating the spatial extent (TAM is out of the considered window). Additionally, three of the eleven spectral channels under investigation are highlighted: IR_108, VIS_008 and WV_073 channels are infrared ($10.8\mu$), visible ($0.8\mu m$) and water vapor ($7.3\mu m$) channels respectively. *Right*: Table summarizing the geographic coordinates and elevation of each station used in the paper.

CrossViViT architecture. In this modified version, the original MLP head is replaced with $N_{heads}$ parallel MLPs, each dedicated to predicting a specific quantile of the distribution for each time step. To achieve this, we employ distinct quantile loss functions for each MLP head. By summing these quantile losses, we obtain a comprehensive Multi-Quantile loss, which serves as the training objective for the model. The quantile loss (Koenker and Hallock, 2001) $L_\alpha(y, \hat{y})$ for the $\alpha$ quantile is defined as:

$$L_\alpha(y, \hat{y}) = \max\{\alpha(\hat{y} - y), (1 - \alpha)(y - \hat{y})\}. \tag{7}$$

The Multi-Quantile loss, aiming to learn multiple quantile predictions $\hat{y}_\alpha$ for a chosen set of quantiles $v_A$, is then defined as: $MQL(y, \hat{y}_{\alpha \in v_A}) = \sum_{\alpha \in v_A} L_\alpha(y, \hat{y}_\alpha)$. The selection of quantile heads $v_\alpha$ is a crucial hyperparameter that determines the density of the output distribution generated by the model. To achieve a 96% prediction interval while maintaining a sufficiently dense distribution, we set the list of quantiles as $v_A = [0.02, 0.1, 0.2, 0.3, 0.4, 0.5, 0.6, 0.7, 0.9, 0.98]$. It is worth noting that, it is possible to assign different weights to the quantiles to guide the learning process. This approach can be beneficial in scenarios where the task requires a preference for overestimation or underestimation. However, in this study, our primary objective was to provide a prediction interval, leading to the conservative choice of quantile distribution and uniformly weighing the $L_\alpha$'s.

## 4   Dataset

This section provides a comprehensive description of the dataset designed for this study and shared publicly, including all the applied pre-processing steps.

### 4.1   Time series

The time-series measurements were obtained from Baseline Surface Radiation Network datasets (Driemel et al., 2018). The experiments in this paper use data from six locations (Figure 2), collected at a 30-minute resolution over a 15-year period (2008-2022). The data captures diverse patterns, ranging from consistent irradiance levels under clear skies to fluctuations caused by intermittent clouds affecting surface solar radiation. The data contains measurements of the pressure in the station, clear sky components, Direct Normal Irradiance (DNI), and Diffuse Horizontal Irradiance (DHI).

The GHI component was computed using the formula: $GHI = DNI \times \cos z + DHI$ where $z$ is the zenith angle of the sun obtained from the pvlib python library (Holmgren et al., 2020). The Ineichen model (Ineichen, 2016) available through the same library is utilized to obtain the clear sky components. The 2008-2016 data is used for training while subsets of the 2017-2022 data and stations are used for validation and test performance evaluations. In this study, the models are trained and evaluated to forecast the GHI component for a 24-hour period ahead, based on a history of 24-hour measurements.

Table 1: Comparison of model performances across **test stations** TAM and CAB, during **test years** (2020-2022) for CAB, and **val years** (2017-2019) for TAM. We report the MAE and RMSE for the easy and difficult splits presented in section 5.2 along with the number of data points for each split. We add the MAE resulting from the Multi-Quantile CrossViViT median prediction, along with $p_t$, the probability for the ground-truth to be included within the interval, averaged across time steps. Additionally, we ablate RoPE against a learned positional embedding.

| Models | Parameters | CAB (2020-2022) | | | | | | TAM (2017-2019) | | | | | |
| --- | --- | --- | --- | --- | --- | --- | --- | --- | --- | --- | --- | --- | --- |
| | | All (9703) | | Easy (5814) | | Hard (3889) | | All (2299) | | Easy (2064) | | Hard (235) | |
| | | MAE | RMSE | MAE | RMSE | MAE | RMSE | MAE | RMSE | MAE | RMSE | MAE | RMSE |
| Persistence | N/A | 63.57 | 131.44 | 52.56 | 109.05 | 80.04 | 159.14 | **32.26** | 94.71 | **20.8** | 59.47 | 132.92 | 238.12 |
| Fourier$_3$ | N/A | 68.91 | 121.23 | 56.51 | 93.854 | 87.46 | 153.29 | 56.0 | 94.85 | 45.48 | 62.62 | 148.42 | 231.47 |
| Fourier$_4$ | N/A | 65.74 | 123.15 | 53.82 | 96.38 | 83.56 | 154.76 | 44.02 | 92.22 | 33.15 | 57.56 | 139.56 | 232.61 |
| Fourier$_5$ | N/A | 64.67 | 124.22 | 52.67 | 97.94 | 82.61 | 155.44 | 40.26 | **91.36** | 28.94 | **55.57** | 139.68 | 233.52 |
| **Clear Sky** (Ineichen, 2016) | N/A | 67.19 | 140.11 | 60.55 | 125.66 | 77.12 | 159.28 | 40.61 | 98.02 | 31.07 | 63.4 | 124.42 | 242.26 |
| **ReFormer** (Kitaev et al., 2020) | 8.6M | 57.42 | 102.73 | 53.75 | 92.97 | 62.92 | 115.81 | 81.6 | 137.04 | 78.57 | 129.72 | 108.22 | 189.55 |
| **Informer** (Zhou et al., 2021) | 56.7M | 72.26 | 122.89 | 70.85 | 118.85 | 74.35 | 128.69 | 83.43 | 140.38 | 82.6 | 138.46 | **90.66** | **156.22** |
| **FiLM** (Zhou et al., 2022b) | 9.4M | 68.37 | 116.86 | 59.66 | 95.35 | 81.4 | 143.11 | 62.72 | 99.71 | 54.99 | 77.63 | 130.66 | 210.58 |
| **PatchTST** (Nie et al., 2023) | 9.6M | 60.76 | 119.41 | 54.77 | 107.71 | 69.7 | 135.01 | 66.94 | 132.44 | 62.4 | 124.25 | 106.77 | 189.72 |
| **LighTS** (Zhang et al., 2022) | 32K | 54.91 | 102.88 | 49.55 | **89.28** | 62.92 | 120.38 | 68.51 | 114.59 | 64.61 | 104.98 | 102.77 | 177.98 |
| **CrossFormer** (Zhang and Yan, 2023) | 227M | 55.98 | 101.84 | 51.59 | 90.2 | 62.55 | 117.11 | 68.85 | 116.45 | 65.4 | 107.88 | 99.16 | 175.08 |
| **FEDFormer** (Zhou et al., 2022a) | 23.6M | 56.38 | 99.27 | 53.08 | 90.13 | 61.31 | 111.54 | 92.12 | 146.52 | 91.13 | 142.83 | 100.82 | 175.64 |
| **DLinear** (Zeng et al., 2022) | 4.7K | 75.01 | 121.01 | 65.21 | 99.72 | 89.65 | 147.21 | 75.54 | 115.40 | 69.04 | 98.74 | 132.67 | 211.28 |
| **AutoFormer** (Wu et al., 2021) | 50.4M | 64.34 | 104.53 | 60.81 | 95.14 | 69.63 | 117.17 | 115.88 | 170.91 | 117.36 | 171.07 | 102.87 | 169.47 |
| **CrossViViT** | 145M | **50.35** | **99.18** | **47.04** | 89.6 | **55.30** | **112.00** | 49.46 | 94.96 | 44.01 | 79.91 | 97.40 | 179.30 |
| **CrossViViT (Learned PE)** [2] | 145M | 51.11 | 103.66 | 47.31 | 95.13 | 56.84 | 115.31 | 109.28 | 196.44 | 111.33 | 197.63 | 91.29 | 185.61 |
| | | MAE | $p_t$ | MAE | $p_t$ | MAE | $p_t$ | MAE | $p_t$ | MAE | $p_t$ | MAE | $p_t$ |
| **Multi-Quantile CrossViViT (small)** | 78.8M | 61.80 | 0.91 | 57.03 | 0.93 | 68.94 | 0.90 | 81.20 | 0.71 | 78.93 | 0.70 | 101.18 | 0.75 |
| **Multi-Quantile CrossViViT (large)** | 145.5M | 74.26 | 0.89 | 68.83 | 0.91 | 82.39 | 0.87 | 79.73 | 0.76 | 76.08 | 0.76 | 111.74 | 0.75 |

## 4.2 Satellite images

In this study, we utilize the EUMETSAT Rapid Scan Service (RSS) dataset (Holmlund, 2003), which spans a period of 15 years from 2008 to 2022, with an original resolution of 5 minutes, later aligned with the time series data. Our analysis focuses on the non-High Resolution Visible (non-HRV) channels, which encompass 11 spectral channels with a spatial resolution of 6-9km and provide comprehensive coverage of the upper third of the Earth, with a particular emphasis on Europe. These channels, including Infrared and Water vapor, offer valuable information for our investigation. To facilitate our analysis, we reprojected the original geostationary projection data onto the World Geodetic System 1984 (WGS 84) coordinate system (Jacob et al., 2022). The region of interest is depicted in Figure 2. Note that one of the 6 stations, TAM, lies slightly outside the region we consider, and therefore represents an out-of-distribution station in term of the context we use.

To augment the contextual information, we computed the optical flow for each channel using the TVL1 algorithm (Sánchez Pérez et al., 2013) from the OpenCV package (Bradski, 2000). The optical flow represents the "velocity" of pixels between consecutive frames, which in our case corresponds to the motion of clouds. Furthermore, we included the elevation map as an additional channel in our dataset. The pre-processed satellite data originally had a resolution of $512^2$, but for computational efficiency, we downscale it to $64^2$.

# 5 Experiments and Results

In this section, we outline the baselines utilized for comparison alongside the suggested architecture. We describe the experimental setup and present the results, benchmarking our framework against state-of-the-art forecasting models across various test configurations. Additionally, we employ a split methodology to assess model performance in challenging prediction scenarios, which hold significant implications for downstream tasks associated with solar irradiance estimation.

The models are trained using a dataset spanning a period of 9 years, from 2008 to 2016, encompassing the stations IZA, CNR, and PAL. Validation is performed on a separate dataset covering 3 years, from 2017 to 2019, for the PAY station. The TAM and CAB stations serve as the test dataset, consisting of 3 years from 2020 to 2022 for CAB, and from 2017 to 2019 for TAM. Each model takes a historical

---

[2]We replace RoPE with learnable *spatial* positional encoding. This was done by adding a learnable parameter $\mathbf{p} \in \mathbb{R}^{N \times d}$ to the encoded input, where $N$ is the number of tokens in each satellite frame.

input of 24 hours and predicts the GHI for the subsequent 24 hours, employing a sliding window approach. Details about the implementation and hyperparameter tuning can be found in the Appendix.

## 5.1 Baselines

We conduct a comprehensive comparison between our approach and several state-of-the-art deep learning architectures specifically designed for forecasting tasks. These architectures are explicitly mentioned in the results tables, such as Table 1. Additionally, we propose *dummy* baselines that are tailored to solar irradiance forecasting, namely:

**Persistence**: This baseline relies solely on the past day's time-series data, considering it as the prediction for future values.

**Clear Sky baseline**: This baseline uses the computable clear sky components of solar irradiance (Ineichen model (Ineichen, 2016)), which represent the total amount of irradiance that would reach the station in the absence of clouds.

**Fourier approximations**: We compute Fourier approximations over the previous day's time series, and apply a low-pass filter to keep a limited number of modes. We consider 3 baselines: $\text{Fourier}_3$, $\text{Fourier}_4$, and $\text{Fourier}_5$, corresponding to approximations with 3, 4, and 5 modes, respectively.

## 5.2 "Hard" vs. "Easy" forecasting scenarios

To assess the capability of the suggested model in capturing cloud-induced variability in GHI forecasts, we perform evaluations on test stations using various time splits, aiming to identify its strengths and limitations in comparison to previous approaches. This analysis allows us to pinpoint the specific scenarios where CrossViViT excels, as well as the areas where it falls short. Furthermore, it provides insights into the comparative strengths and weaknesses of previous approaches.

Given the favorable performance of the Persistence baseline when the GHI values exhibit similarity between consecutive days, we propose a time split approach that categorizes examples as either "Easy" or "Hard" based on the extent of GHI variation. The "Easy" examples entail minimal changes in GHI across consecutive days (the Persistence baseline works well), while the "Hard" examples involve significant variations (the Persistence baseline fails). To quantify the similarity, we employ a measure based on the ratio of the area under the GHI curve for the two days. In order to assign equal importance to ratios such as 0.5 (indicating GHI half that of the previous day) and 2 (indicating GHI double that of the previous day), we utilize the measure $r = \left| \log \frac{y}{y_{prev}} \right|$. Here, $y$ represents the GHI over a 24-hour period, and $y_{\text{prev}}$ represents the GHI over the previous 24 hours. Accordingly, we categorize cases as "Easy" when $r < \left| \log \left( \frac{2}{3} \right) \right|$, and "Hard" otherwise.

## 5.3 Performance on stations and years outside the training distribution

Table 1 presents a comprehensive comparison of the proposed CrossViViT approach with state-of-the-art timeseries models and *dummy* baselines. The evaluation is conducted on the test stations TAM and CAB, covering the periods 2017-2019 and 2020-2022, respectively. It is important to note that due to the unavailability of data for TAM during the 2020-2022 period, we perform the evaluation using the 2017-2019 data instead.

During the 2017-2019 period, CrossViViT achieves the lowest MAE compared to the time-series models on the TAM station. However, it is important to note that the persistence baseline still outperforms our approach. This performance disparity can be attributed to the characteristics of the TAM station, which is situated in a desert region characterized by predominantly clear and sunny days. As we incorporate cloud information, it may occasionally underestimate the GHI in such clear-sky conditions. Furthermore, the training dataset consists of data from only one "sunny" station located in the Canary Islands (IZA), limiting the availability of examples to effectively learn clear-sky patterns. Based on these results, one can see that for stations characterized by low irradiance intermittency, a combination of persistence and clear-sky models might be sufficient. For the 2020-2022 period on the CAB station, CrossViViT outperforms all baselines across different time splits. This notable improvement can be attributed to the specific meteorological conditions of the CAB station, which experiences a higher frequency of cloudy days. This aligns with the primary focus of our research, which aims to accurately forecast GHI under cloudier conditions. Predictions visualisations can be

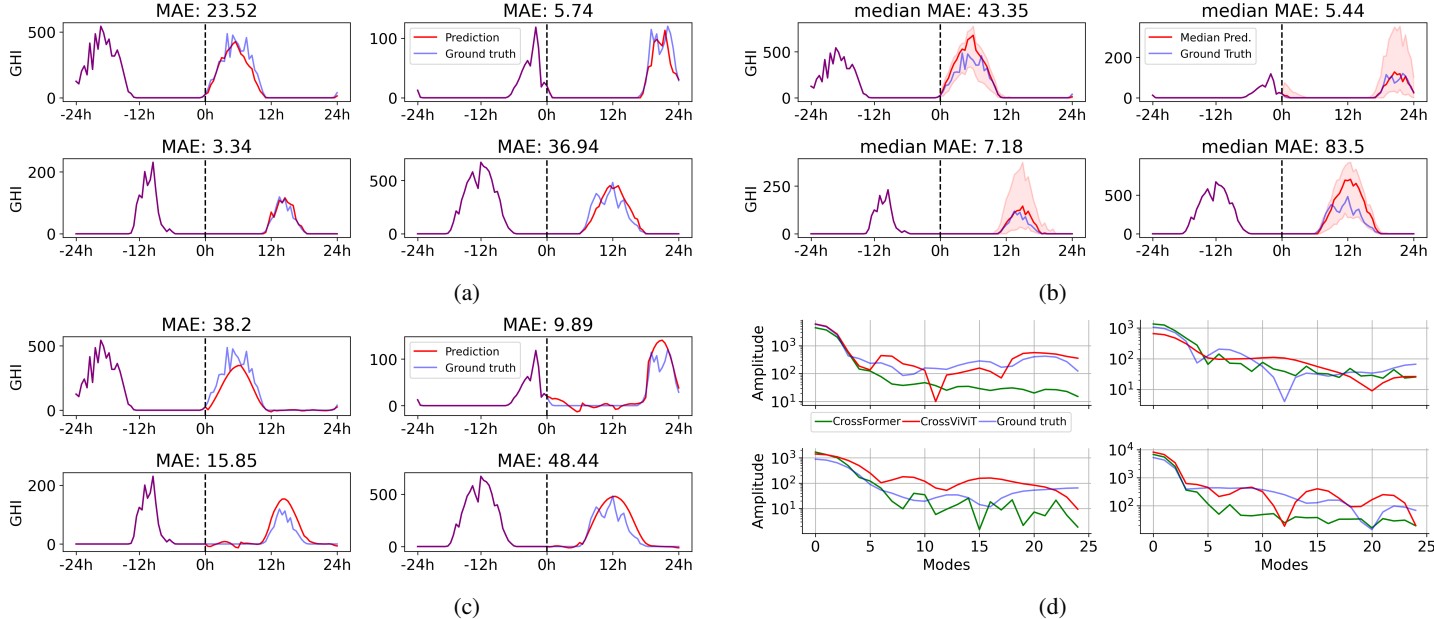

Figure 3: **Prediction visualisations from CrossViViT for four examples in CAB station, on the 2020-2022 test period.** (a) CrossViViT predictions. (b) Multi-Quantile CrossViViT median ($q_{0.50}$ quantile) predictions with $[q_{0.02}, q_{0.98}]$ prediction interval. (c) Predictions from a strong baseline, CrossFormer, (d) Fourier spectrum of the target, our prediction, and CrossFormer prediction. Figure (a) illustrates that CrossViViT closely aligns with the ground truth by effectively capturing cloud variations, whereas CrossFormer assumes a clear-sky pattern. This is confirmed by the Fourier spectra depicted in (d), where CrossFormer's spectrum exhibits a rapid decay in contrast to CrossViViT.

seen in Figure 3, along with the comparison of the fourier spectra of our prediction, the ground truth and a strong baseline, CrossFormer.

### 5.4 Zero-shot forecasting on unseen stations and in-distribution years

To evaluate the zero-shot capabilities of CrossViViT in terms of performance on unseen stations, we maintain the **same time period as the training data**, thereby emphasizing the **spatial dimension** of the analysis. The radar plots presented in Figure 4 demonstrate that our approach consistently outperforms the baselines across all splits for the CAB and PAY stations, which are characterized by higher cloud cover. However, it is noteworthy that Persistence remains competitive for the TAM station, particularly in the "easy" splits where irradiance variations are minimal throughout the day. This observation further underscores the efficacy of CrossViViT in accurately accounting for cloud conditions, particularly when examining the performance metrics of the "Hard" split.

### 5.5 Multi-Quantile results

Table 1 also shows results of the Multi-Quantile CrossViViT, including the MAE of the median prediction (the 0.5 quantile), along with the test confidence $p_t$ obtained for the prediction interval: the probability for the ground-truth to be included within the interval, for each time step, averaged across the entire dataset considered. As for the other models, the evaluation is conducted on the test stations TAM and CAB, covering the periods 2017-2019 and 2020-2022, respectively. It is important to note that for this Multi-Quantile version, the goal is not to provide the best prediction from the median but rather to provide confident prediction intervals, with a high $p_t$, ideally close to 96%, which we theoretically should reach using 0.02 and 0.98 extreme quantiles. We include a small version of the Multi-Quantile model, and a large one, matching the number of parameters used for our Cross ViViT model. The small model provides the best performance.

The prediction intervals achieved a high level of confidence, surpassing 0.9, for the unseen CAB station. However, for the TAM station, the harsh environmental conditions of the desert posed a

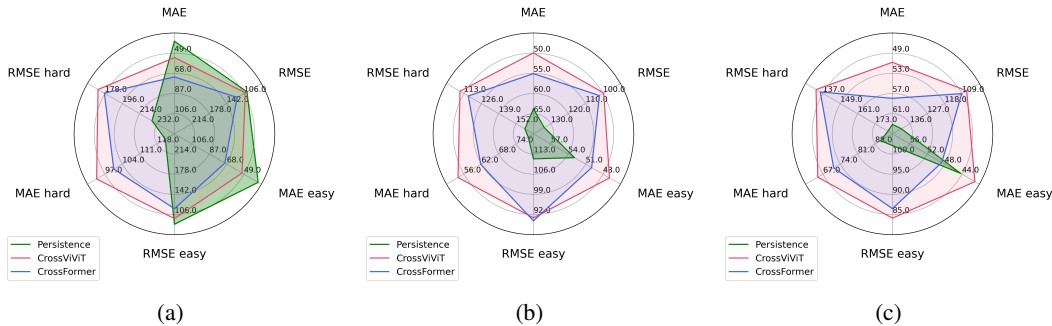

Figure 4: Radar plots illustrating the comparative analysis of CrossViViT performance with Persistence and CrossFormer models, during **training years** (the in-distribution period from 2008 to 2016), on the **unobserved stations** of TAM, CAB, and PAY, respectively on (a), (b) and (c). CrossViViT demonstrates greater versatility compared to CrossFormer in forecasting under diverse conditions. Persistence struggles to accurately predict under challenging conditions, particularly on "Hard" days.

challenge for reliable estimation of prediction intervals, due to the abundance of clear sky components which is not seen during training. Although the median prediction results were comparatively inferior to those of a baseline method, it demonstrates consistent patterns in its variation across easy and difficult cases. Furthermore, the test confidence of the proposed method remains relatively constant across different splits, for both stations. Prediction visualisations can be seen on Figure 3.

## 5.6 Ablating the Rotary Positional Embedding

In order to assess the influence of RoPE, we ablate this design choice against a learned positional encoding which was the standard encoding used in ViViT. As Table 1 shows, RoPE contributes to the good performance in OOD stations and OOD years. Moreover, in the hard cases of the CAB station, RoPE outperforms the learnable positional encoding showcasing its importance in capturing cloud-induced variations.

## 6 Conclusion, Limitations, and Future Work

We present CrossViViT, an architecture for accurate day-ahead time-series forecasting of solar irradiance using the spatio-temporal context derived from satellite data. We suggest a testing scheme that captures model performance in crucial situations, such as days with varying cloudy conditions, and we enable the extraction of a distribution for each time step prediction. We also introduce a new multi-modal dataset that provides a diverse collection of solar irradiance and related physical variables from multiple geographically diverse solar stations. CrossViViT exhibits robust performance in solar irradiance forecasting, including zero-shot tests at unobserved solar stations during unobserved years, and holds great promise in promoting the effective integration of solar power into the grid.

However, there are some limitations to our study. Firstly, we would benefit to include more test years and stations to further validate the effectiveness of our approach. It would also be interesting to explore different past and future horizons to further evaluate the robustness of our model across different prediction and context horizons. The training time remains our key limitation since it takes close to 5 hours per epoch on a single GPU (see Appendix), but we plan on optimizing the architecture further in future work. Lastly, we plan to investigate the use of a cropping methodology on the context as a regularization method. Despite these limitations, our study clearly highlights the importance of incorporating spatio-temporal context in solar irradiance forecasting and demonstrates the potential of deep learning in addressing the challenges associated with variability in solar irradiance. Given the promising results observed in our current study, we intend to investigate the applicability of CrossViViT to other physical variables that are significantly influenced by their surrounding context.

## Acknowledgments and Disclosure of Funding

This research has been funded by the Government of Quebec, with additional computational resources supplied by Mila (mila.quebec). It is conducted as part of an international partnership project between Mila, Moroccan agency for sustainable energy (Masen), l'UQAR, Polytechnique Montreal, l'Ecole Mohammadia d'Ingénieurs, and l'Institut de Valorisation des Données (IVADO). Our special thanks to MASEN R&D team for their precious collaboration. We extend our sincere appreciation to Jacob Bieker from Open Climate Fix, whose facilitation in accessing EUMETSAT data, coupled with his invaluable assistance regarding data availability, markedly enriched our study.

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

# A Experimental details

In this section, we present details about training CrossViViT and its Multi-Quantile variant, as well as the time-series baselines.

## A.1 Training setup

### A.1.1 CrossViViT

CrossViViT integrates two modalities: satellite video data and time-series station data. For both modalities, essential information such as geographic coordinates, elevation, and precise time-stamps is available. In this section, we provide a comprehensive explanation of the encoding process for each feature and conclude by presenting the hyperparameters of the model.

We first start by encoding the timestamps. For each time point, we have access to the following *time features*: The year, the month, the day, the hour and the minute at which the measurement was made. We use a cyclical embedding to encode these time features discarding the year. For a time feature $x$, its corresponding embedding can be expressed as:

$$\left[ \sin\left( \frac{2\pi x}{\omega(x)} \right), \cos\left( \frac{2\pi x}{\omega(x)} \right) \right] \tag{8}$$

Where $\omega(x)$ is the frequency for time feature $x$ (see Table 2) for the frequency of each time feature). We concatenate these features to form our final time embedding that we simply concatenate to the context and the time-series channels respectively. Furthermore, we incorporate the elevation data for each coordinate in both the context and the time-series. Specifically, the elevation values are concatenated to their corresponding channels in the context and time-series representations.

Regarding the geographic coordinates, we possess information regarding the latitude and longitude for both the context and the station. These coordinates are normalized so as to lie in $[-1, 1]$:

$$\begin{cases} \text{lat} \leftarrow 2\left( \frac{\text{lat}+90}{180} \right) - 1 \\ \text{lon} \leftarrow 2\left( \frac{\text{lon}+180}{360} \right) - 1 \end{cases} \tag{9}$$

Table 2: Frequency of each time feature.

| Time feature | Frequency |
|:---:|:---:|
| Month | 12 |
| Day | 31 |
| Hour | 24 |
| Minute | 60 |

**Rotary Positional Emebedding**   Next, we embed these coordinates using Rotary Positional Embedding (RoPE) that we provide a PyTorch implementation for:

```
     Rotary Positional Embedding
 1   class AxialRotaryEmbedding(nn.Module):
 2       def __init__(self, dim, max_freq):
 3           super().__init__()
 4           self.dim = dim
 5           scales = torch.linspace(1.0, max_freq / 2, dim // 4)
 6
 7           self.register_buffer("scales", scales)
 8
 9       def forward(self, coords: torch.Tensor):
10           """
11           Args:
12               coords (torch.Tensor): Coordinates of shape [B, 2, height, width]
13           """
14           seq_x = coords[:, 0, 0, :]
15           seq_x = seq_x.unsqueeze(-1)
16           seq_y = coords[:, 1, :, 0]
17           seq_y = seq_y.unsqueeze(-1)
18
19           scales = self.scales[(*((None, None)), Ellipsis)]
20           scales = scales.to(coords)
21
22           seq_x = seq_x * scales * pi
23           seq_y = seq_y * scales * pi
24
25           x_sinu = repeat(seq_x, "b i d -> b i j d", j=seq_y.shape[1])
26           y_sinu = repeat(seq_y, "b j d -> b i j d", i=seq_x.shape[1])
27
28           sin = torch.cat((x_sinu.sin(), y_sinu.sin()), dim=-1)
29           cos = torch.cat((x_sinu.cos(), y_sinu.cos()), dim=-1)
30
31           sin, cos = map(lambda t: rearrange(t, "b i j d -> b (i j) d"), (sin,
             ↪ cos))
32           sin, cos = map(lambda t: repeat(t, "b n d -> b n (d j)", j=2), (sin,
             ↪ cos))
33           return sin, cos
```

**Training configuration**   CrossViViT was trained on two RTX8000 GPUs, over 17 epochs with early stopping. Its Multi-Quantile variant was also trained on two RTX8000 GPUs, over 12 epochs with early stopping. The remaining settings are identical for both variants: An effective batch size of 20 was utilized for both models; The training process employed the AdamW optimizer (Loshchilov and Hutter, 2019) with a weight decay of 0.05; A cosine warmup strategy was implemented, gradually increasing the learning rate from 0 to 0.0016 over five epochs before starting the decay phase.

Below, we highlight the relevant model hyperparameters for CrossViViT and Multi-Quantile Cross-ViViT:



**CrossViViT hyperparameters**

```
1  patch_size: [8, 8]
2  use_glu: True
3  max_freq: 128
4  num_mlp_heads: 1
5
6  ctx_masking_ratio: 0.99
7  ts_masking_ratio: 0
8
9  # These hyperparameters apply to the encoding transformers and cross-attention
10 dim: 384
11 depth: 16

12 heads: 12
13 mlp_ratio: 4
14 dim_head: 64
15 dropout: 0.4
16
17 # These only apply to the decoding transformer
18 decoder_dim: 128
19 decoder_depth: 4
20 decoder_heads: 6
21 decoder_dim_head: 128
```




**Multi-Quantile CrossViViT hyperparameters**

```
1  patch_size: [8, 8]
2  use_glu: True
3  max_freq: 128
4  num_mlp_heads: 11
5  quantiles: [0.02, 0.1, 0.2, 0.3, 0.4, 0.5, 0.6, 0.7, 0.8, 0.9, 0.98]
6
7  ctx_masking_ratio: 0.99
8  ts_masking_ratio: 0
9
10 # These hyperparameters apply to the encoding transformers and cross-attention
11 dim: 256

12 depth: 16
13 heads: 12
14 mlp_ratio: 4
15 dim_head: 64
16 dropout: 0.4
17
18 # These only apply to the decoding transformer
19 decoder_dim: 128
20 decoder_depth: 4
21 decoder_heads: 6
22 decoder_dim_head: 128
```


### A.1.2 Time-series baselines

We conducted training on a total of nine baseline models, and we emphasize the importance of the hyperparameters used for each of these models. Below, we provide a comprehensive overview of the hyperparameters employed in our study:

- **seq_len**: Input sequence length.
- **label_len**: Start token length.
- **pred_len**: Prediction sequence length.
- **enc_in**: Encoder input size.

- **dec_in**: Decoder input size.
- **e_layers**: Number of encoder layers.
- **d_layers**: Number of decoder layers.
- **c_out**: Output size.
- **d_model**: Dimension of model.
- **n_heads**: Number of attention heads.
- **d_ff**: Dimension of Fully Connected Network.
- **factor**: Attention factor.
- **embed**: Time features encoding, options:[timeF, fixed, learned].
- **distil**: Whether to use distilling in encoder.
- **moving average**: Window size of moving average kernel.

We adapted the majority of the baselines using the Time Series Library (TSlib (Wu et al., 2023)), which served as a valuable resource in our experimentation. We refer the reader to the original papers, which served as a base for the hyperparameters utilized in our study, in order to have a comprehensive understanding of the models and the training settings.

**LightTS**
```
1  model:
2    enc_in: 10
3    seq_len: 48
4    pred_len: 48
5    d_model: 256
6    dropout: 0.05
7    chunk_size: 24
```

**FiLM**
```
1  model:
2    enc_in: 10
3    seq_len: 48
4    label_len: 24
5    pred_len: 48
6    e_layers: 2
7    ratio: 0.4
```

**DLinear**
```
1  model:
2    enc_in: 10
3    seq_len: 48
4    pred_len: 48
5    moving_avg: 25
6    individual: False
```

**Crossformer**
```
1   model:
2     enc_in: 10
3     seq_len: 48
4     pred_len: 48
5     d_model: 1024
6     n_heads: 2
7     e_layers: 4
8     d_ff: 2048
9     factor: 10
10    dropout: 0.01
```

**Reformer**
```
1   model:
2     enc_in: 10
3     c_out: 1
4     seq_len: 48
5     pred_len: 48
6     d_model: 512
7     n_heads: 8
8     e_layers: 3
9     d_ff: 2048
10    factor: 5
11    dropout: 0.05
12    embed: timeF
13    activation: gelu
```

**PatchTST**
```
1   model:
2     enc_in: 10
3     c_out: 1
4     seq_len: 48
5     pred_len: 48
6     d_model: 1024
7     n_heads: 6
8     e_layers: 3
9     d_ff: 2048
10    factor: 10
11    dropout: 0.05
12    embed: timeF
13    activation: gelu
```

```
Autoformer
1  model:
2    enc_in: 10
3    dec_in: 10
4    c_out:
5    seq_len: 48
6    label_len: 24
7    pred_len: 48
8    moving_avg: 25
9    d_model: 1024
10   n_heads: 8
11   e_layers: 3
12   d_layers: 2
13   d_ff: 2048
14   factor: 10
15   dropout: 0.01
16   embed: timeF
17   activation: gelu
```

```
Informer
1  model:
2    enc_in: 10
3    dec_in: 10
4    c_out: 1
5    label_len: 24
6    pred_len: 48
7    d_model: 2048
8    n_heads: 4
9    e_layers: 2
10   d_layers: 2
11   d_ff: 2048
12   factor: 5
13   dropout: 0.1
14   embed: timeF
15   activation: gelu
16   distil: True
```

```
FEDformer
1  model:
2    enc_in: 10
3    dec_in: 10
4    c_out: 1
5    seq_len: 48
6    label_len: 24
7    pred_len: 48
8    moving_avg: 25
9    d_model: 512
10   n_heads: 8
11   e_layers: 3
12   d_layers: 2
13   d_ff: 2048
14   dropout: 0.05
15   version: fourier
16   mode_select: random
17   modes: 32
```

The training of the baselines took place on a single RTX8000 GPU over the course of 100 epochs. During training, a batch size of 64 was consistently employed. For model optimization, we utilized the AdamW optimizer (Loshchilov and Hutter, 2019), incorporating a weight decay value set to 0.05. Moreover, we implemented a learning rate reduction strategy known as Reduce Learning Rate on Plateau, which gradually decreased the learning rate by a factor of 0.5 after a patience of 10 epochs.

For the hyperparameter tuning of the baselines, we employed the Orion package (Bouthillier et al., 2022), which is an asynchronous framework designed for black-box function optimization. Its primary objective is to serve as a meta-optimizer specifically tailored for machine learning models. As an example, we present the details of the Crossformer hyperparameter tuning scheme, showcasing the approach we followed to optimize its performance:

```
Crossformer hyperparameters tuning
1  params:
2    optimizer.lr: loguniform(1e-8, 0.1)
3    optimizer.weight_decay: loguniform(1e-10, 1)
4    d_model: choices([128,256,512,1024,2048])
5    d_ff: choices([1024,2048,4096])
6    n_heads: choices([1,2,4,8,16])
7    e_layers: choices([1,2,3,4,5])
8    dropout: choices([0.01,0.05,0.1,0.2,0.25])
9    factor: choices([2,5,10])
10   max_epochs: fidelity(low=5, high=100, base=3)
```

## B    Additional results and visualisations

### B.1    Performance on (unobserved) validation PAY station, on validation and test years

This section presents a comprehensive comparative analysis that assesses the performance of Cross-ViViT in relation to various timeseries approaches and solar irradiance baselines specifically for the PAY station. The evaluation encompasses both the validation period (2017-2019) and the test period (2020-2022), with the validation period serving as the basis for model selection. The results, as presented in Table 3, offer compelling evidence of the superior performance of CrossViViT compared to the alternative approaches across all evaluation splits. These findings underscore the crucial role of accurately capturing cloud dynamics in solar irradiance forecasting, which is particularly pronounced in the "Hard" splits.

Table 3: Comparison of model performances in the **val station** PAY, during **test years** (2020-2022) and **val years** (2017-2019). We report the MAE and RMSE for the easy and difficult splits (presented in the main paper) along with the number of data points for each split. We add the MAE resulting from the Multi-Quantile CrossViViT median prediction, along with $p_t$, the probability for the ground-truth to be included within the interval, averaged across time steps.

| Models | Parameters | PAY (Test years - 2020-2022) | | | | | | PAY (Val years - 2017-2019) | | | | | |
| | | All (12171) | | Easy (8053) | | Hard (4118) | | All (27166) | | Easy (16683) | | Hard (10483) | |
| | | MAE | RMSE | MAE | RMSE | MAE | RMSE | MAE | RMSE | MAE | RMSE | MAE | RMSE |
| Persistence | N/A | 61.9 | 140.05 | 44.98 | 105.47 | 94.99 | 190.31 | 63.61 | 141.16 | 48.57 | 108.91 | 87.54 | 180.99 |
| Fourier₃ | N/A | 69.21 | 129.29 | 52.17 | 91.61 | 102.53 | 181.63 | 71.71 | 130.58 | 56.03 | 94.81 | 96.68 | 172.86 |
| Fourier₄ | N/A | 65.67 | 131.13 | 48.77 | 93.68 | 98.72 | 183.48 | 67.35 | 132.41 | 51.8 | 96.71 | 92.08 | 174.79 |
| Fourier₅ | N/A | 63.39 | 132.38 | 46.14 | 95.08 | 97.14 | 184.71 | 65.71 | 133.87 | 50.03 | 98.61 | 90.66 | 175.98 |
| Clear Sky (Ineichen, 2016) | N/A | 65.99 | 138.78 | 56.41 | 116.72 | 84.72 | 174.03 | 62.12 | 131.69 | 52.59 | 109.91 | 77.28 | 160.36 |
| ReFormer (Kitaev et al., 2020) | 8.6M | 59.15 | 109.75 | 51.67 | 91.67 | 73.79 | 138.44 | 59.3 | 109.3 | 54.06 | 94.31 | 67.64 | 129.62 |
| Informer (Zhou et al., 2021) | 56.7M | 79.18 | 133.47 | 75.8 | 126.07 | 85.78 | 146.86 | 77.51 | 131.72 | 75.65 | 126.58 | 80.46 | 139.52 |
| FiLM (Zhou et al., 2022b) | 9.4M | 69.87 | 124.86 | 55.93 | 91.15 | 97.11 | 172.72 | 71.1 | 125.32 | 58.35 | 94.17 | 91.39 | 163.06 |
| PatchTST (Nie et al., 2023) | 9.6M | 63.21 | 131.64 | 53.06 | 113.78 | 83.06 | 160.93 | 63.88 | 131.14 | 55.47 | 115.59 | 77.25 | 152.64 |
| LighTS (Zhang et al., 2022) | 32K | 58.57 | 111.22 | 48.93 | 88.69 | 77.4 | 145.53 | 58.22 | 110.84 | 50.72 | 91.24 | 70.16 | 136.33 |
| CrossFormer (Zhang and Yan, 2023) | 227M | 59.64 | 111.03 | 51.04 | 90.34 | 76.45 | 143.08 | 59.39 | 111.49 | 52.55 | 93.15 | 70.29 | 135.66 |
| FEDFormer (Zhou et al., 2022a) | 23.6M | 63.44 | 110.62 | 58.15 | 97.33 | 73.79 | **132.83** | 62.46 | 109.92 | 59.88 | 100.61 | 66.57 | **123.3** |
| DLinear (Zeng et al., 2022) | 4.7K | 75.09 | 128.64 | 59.42 | 93.82 | 105.74 | 178.04 | 76.78 | 129.45 | 62.64 | 97.93 | 99.29 | 167.81 |
| AutoFormer (Wu et al., 2021) | 50.4M | 73.36 | 117.22 | 68.89 | 105.52 | 82.12 | 137.25 | 71.39 | 114.96 | 69.42 | 106.79 | 74.54 | 126.88 |
| **CrossViViT** | 145M | **51.47** | **107.73** | **41.59** | **85.14** | **70.81** | 141.87 | **52.02** | **108.77** | **44.71** | **90.47** | **63.65** | 132.79 |
| | | MAE | $p_t$ | MAE | $p_t$ | MAE | $p_t$ | MAE | $p_t$ | MAE | $p_t$ | MAE | $p_t$ |
| **Multi-Quantile CrossViViT (small)** | 78.8M | 59.06 | 0.83 | 54.96 | 0.83 | 67.06 | 0.84 | 57.65 | 0.86 | 51.75 | 0.87 | 67.05 | 0.86 |
| **Multi-Quantile CrossViViT (large)** | 145.5M | 74.18 | 0.89 | 68.69 | 0.91 | 82.38 | 0.87 | 59.97 | 0.87 | 52.67 | 0.88 | 71.59 | 0.85 |

## B.2 Performance in the operational setting: 00:00 to 23:00 window only

In this section, we present an assessment of CrossViViT's performance in an operational context, specifically focusing on evaluations conducted within the 00:00 to 23:00 time window. This approach provides insights into the model's effectiveness in its intended real-world application, where it is typically executed once daily, precisely at midnight. This evaluation setup offers a more accurate representation of the model's practical utility compared to the sliding window approach. Table 4 shows that the relative performance of CrossViViT over baselines is still maintained which demonstrates the usefulness of our approach in real-world scenarios.

Table 4: Comparison of model performances in the **test station** CAB, during **test years** (2020-2022) and **test station** TAM on **val years** (2017-2019) for 00:00 to 23:00 windows only. We report the MAE and RMSE for the easy and hard splits along with the number of data points for each split.

| Models | Parameters | CAB (2020-2022) | | | | | | TAM (2017-2019) | | | | | |
| | | All (207) | | Easy (120) | | Hard (87) | | All (48) | | Easy (42) | | Hard (6) | |
| | | MAE | RMSE | MAE | RMSE | MAE | RMSE | MAE | RMSE | MAE | RMSE | MAE | RMSE |
| Persistence | N/A | 63.19 | 130.5 | 49.91 | 103.23 | 81.49 | 160.7 | **33.94** | 98.15 | **19.75** | 53.91 | 133.32 | 238.15 |
| Fourier₃ | N/A | 69.49 | 122.68 | 54.9 | 90.38 | 89.63 | 156.67 | 57.7 | 99.09 | 44.57 | 58.76 | 149.62 | 233.2 |
| Fourier₄ | N/A | 65.78 | 123.84 | 51.71 | 92.03 | 85.19 | 157.5 | 45.55 | 95.98 | 31.95 | 52.41 | 140.79 | 233.4 |
| Fourier₅ | N/A | 64.55 | 124.51 | 50.24 | 93.08 | 84.3 | 157.9 | 41.91 | **94.95** | 27.79 | **49.58** | 140.77 | 234.33 |
| Clear Sky (Ineichen, 2016) | N/A | 67.31 | 140.42 | 58.68 | 121.49 | 79.21 | 162.96 | 40.93 | 99.03 | 29.42 | 55.82 | 121.5 | 237.99 |
| ReFormer (Kitaev et al., 2020) | 8.6M | 60.64 | 104.52 | 55.98 | 92.28 | 67.07 | 119.35 | 81.5 | 134.89 | 77.95 | 126.06 | 106.36 | 185.29 |
| Informer (Zhou et al., 2021) | 56.7M | 72.82 | 125.44 | 69.23 | 118.1 | 77.78 | 134.91 | 84.51 | 137.51 | 81.91 | 131.83 | 102.74 | **172.11** |
| FiLM (Zhou et al., 2022b) | 9.4M | 69.36 | 120.87 | 58.65 | 95.01 | 84.14 | 149.36 | 63.42 | 105.46 | 53.4 | 77.02 | 133.58 | 217.82 |
| PatchTST (Nie et al., 2023) | 9.6M | 64.37 | 124.09 | 54.56 | 108.37 | 74.84 | 142.96 | 73.16 | 142.01 | 67.18 | 130.89 | 115.03 | 203.49 |
| LighTS (Zhang et al., 2022) | 32K | 61.54 | 105.14 | 54.56 | 87.22 | 71.18 | 125.73 | 61.73 | 101.48 | 52.95 | 80.29 | 123.19 | 193.04 |
| CrossFormer (Zhang and Yan, 2023) | 227M | 58.19 | 104.66 | 52.96 | 90.03 | 65.4 | 121.99 | 66.25 | 113.67 | 58.81 | 95.04 | 118.33 | 200.37 |
| FEDFormer (Zhou et al., 2022a) | 23.6M | 59.11 | 104.56 | 52.88 | 88.58 | 67.71 | 123.25 | 114.47 | 176.24 | 116.59 | 176.21 | 99.62 | 176.42 |
| DLinear (Zeng et al., 2022) | 4.7K | 77.36 | 125.17 | 65.61 | 100.19 | 93.57 | 153.08 | 79.36 | 121.69 | 70.71 | 99.87 | 139.9 | 220.55 |
| AutoFormer (Wu et al., 2021) | 50.4M | 72.16 | 110.54 | 67.87 | 97.6 | 78.07 | 126.23 | 141.97 | 204.4 | 146.02 | 208.11 | 113.62 | 176.2 |
| **CrossViViT** | 145M | **51.91** | **101.58** | **46.24** | **87** | **59.73** | **118.8** | 52.72 | 99.11 | 46.47 | 82.53 | **96.42** | 175.79 |

## B.3 Performance evaluation with MAPE

In this section, we report the Mean Absolute Percentage Error (MAPE)[3] on test station CAB during test years (2020-2022) and test station TAM during val years (2017-2019) (see Table 5). The MAPE highlights even more the strength of our approach compared to the baselines. We also present results

---

[3]Since during the night, the GHI values are zero, we clamp the denominator by $\epsilon = 1$ since the precision of the GHI values in the dataset is 1.

Table 5: Comparison of model performances across **test stations** TAM and CAB, during **test years** (2020-2022) for CAB, and **val years** (2017-2019) for TAM, using the **Mean Absolute Percentage Error (MAPE)** metric. We report the MAPE for the easy and difficult splits presented in section 5.2 along with the number of data points for each split. We add the MAPE resulting from the Multi-Quantile CrossViViT median prediction, along with $p_t$, the probability for the ground-truth to be included within the interval, averaged across time steps. Additionally, we ablate RoPE against a learned positional embedding.

| Models | Parameters | CAB (2020-2022) | | | TAM (2017-2019) | | |
|---|---|---|---|---|---|---|---|
| | | All (9703) | Easy (5814) | Hard (3889) | All (2299) | Easy (2064) | Hard (235) |
| **Persistence** | N/A | 0.54 | 0.37 | 0.8 | **0.14** | **0.08** | 0.67 |
| **Fourier$_3$** | N/A | 8.43 | 8.22 | 8.76 | 19.98 | 20.37 | 16.6 |
| **Fourier$_4$** | N/A | 5.42 | 5.58 | 5.18 | 7.54 | 7.4 | 8.83 |
| **Fourier$_5$** | N/A | 4.21 | 4.17 | 4.26 | 7.55 | 7.44 | 8.47 |
| **Clear Sky** (Ineichen, 2016) | N/A | 0.72 | 0.53 | 1.02 | 0.28 | 0.21 | 0.9 |
| **ReFormer** (Kitaev et al., 2020) | 8.6M | 4.05 | 3.7 | 4.59 | 5.88 | 5.89 | 5.78 |
| **Informer** (Zhou et al., 2021) | 56.7M | 6.12 | 4.92 | 7.93 | 5.44 | 5.42 | 5.57 |
| **FiLM** (Zhou et al., 2022b) | 9.4M | 8.88 | 9.18 | 8.45 | 11.62 | 11.64 | 11.43 |
| **PatchTST** (Nie et al., 2023) | 9.6M | 2.53 | 2.4 | 2.74 | 3.32 | 3.29 | 3.59 |
| **LighTS** (Zhang et al., 2022) | 32K | 7.08 | 7.08 | 7.08 | 12.28 | 11.82 | 16.29 |
| **CrossFormer** (Zhang and Yan, 2023) | 227M | 3.07 | 2.76 | 3.54 | 3.57 | 3.54 | 3.86 |
| **FEDFormer** (Zhou et al., 2022a) | 23.6M | 5.35 | 4.95 | 5.96 | 12.83 | 13.15 | 10.01 |
| **DLinear** (Zeng et al., 2022) | 4.7K | 13.98 | 12.99 | 16.37 | 13.11 | 12.98 | 14.21 |
| **AutoFormer** (Wu et al., 2021) | 50.4M | 10.47 | 9.77 | 11.51 | 23.33 | 24.18 | 15.91 |
| **CrossViViT** | 145M | **0.43** | **0.33** | 0.58 | 0.17 | 0.14 | **0.47** |
| **CrossViViT (Learned PE)**[4] | 145M | 0.83 | 0.64 | 1.13 | 0.95 | 0.99 | 0.6 |

| | | MAPE | $p_t$ | MAPE | $p_t$ | MAPE | $p_t$ | MAPE | $p_t$ | MAPE | $p_t$ | MAPE | $p_t$ |
|---|---|---|---|---|---|---|---|---|---|---|---|---|---|
| **Multi-Quantile CrossViViT (small)** | 78.8M | 0.61 | 0.92 | 0.43 | 0.93 | 0.87 | 0.90 | 0.61 | 0.92 | 0.44 | 0.93 | 0.87 | 0.90 |
| **Multi-Quantile CrossViViT (large)** | 145.5M | 0.69 | 0.89 | 0.49 | 0.91 | 0.99 | 0.87 | 0.69 | 0.89 | 0.49 | 0.91 | 0.99 | 0.87 |

of the ablated model for which we replace RoPE with a traditional learned Positional Encoding. Despite showing a better performance than most baselines, it does showcase a lower performance than the original CrossViViT, which underlines the importance of the choice of RoPE.

## B.4 Inference times

Table 6 shows inference time metrics of CrossViViT and the time-series baselines and we do so by reporting two main metrics:

- **Latency**: The time it takes for the model to process one instance (batch size=1).
- **MAC**: Number of multiply-accumulate operations (MAC). A multiply-accumulate operation corresponds to $a + (b \times c)$ which counts as one operation.

The above metrics have all been computed on a single RTX8000 GPU. From Table 6, we can see that CrossViViT and its multi-quantile variant have the longest inference and training times which is normal given that we incorporate two different modalities, one of which is usually very expensive to process (i.e. satellite data). But we argue that in an operational setting, CrossViViT only needs to be executed once a day which means that while it is desirable to have a relatively short inference time, it may not be of paramount importance, especially if such speed gains come at the expense of predictive accuracy. It is worth acknowledging that a noteworthy limitation lies in the training time, which could potentially be improved through optimization efforts in future work.

## B.5 Visualisations for day ahead time series predictions

This section presents visualizations of predictions generated by CrossViViT and CrossFormer on the PAY station for both the validation period (2017-2019) (see Figure 6) and the test period (2020-2022) (see Figure 5). We also present visualizations of predictions generated by the Multi-Quantile version, for the two periods, on the PAY station. The predictions depicted in red are the median (50% quantile) estimation from the model, and the generated prediction interval is defined as the interval between the two predefined quantiles: $[q_{0.02}, q_{0.98}]$

---

[4]We replace RoPE with learnable *spatial* positional encoding. This was done by adding a learnable parameter $\mathbf{p} \in \mathbb{R}^{N \times d}$ to the encoded input, where $N$ is the number of tokens in each satellite frame.

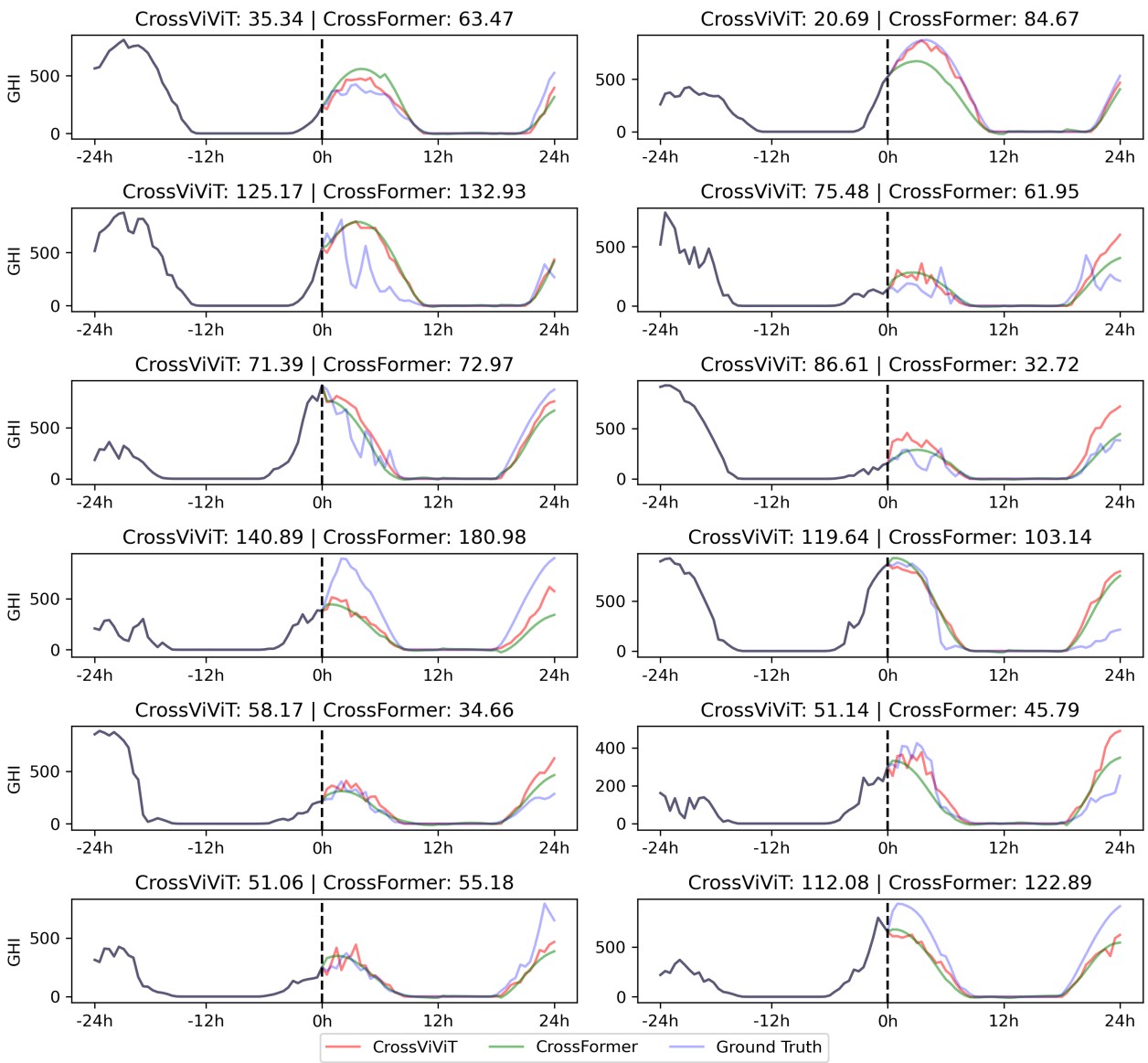

Figure 5: Visualizations of CrossViViT and CrossFormer are presented for a subset of 12 randomly selected days from the "Hard" split on the PAY station during the test period spanning from 2020 to 2022.

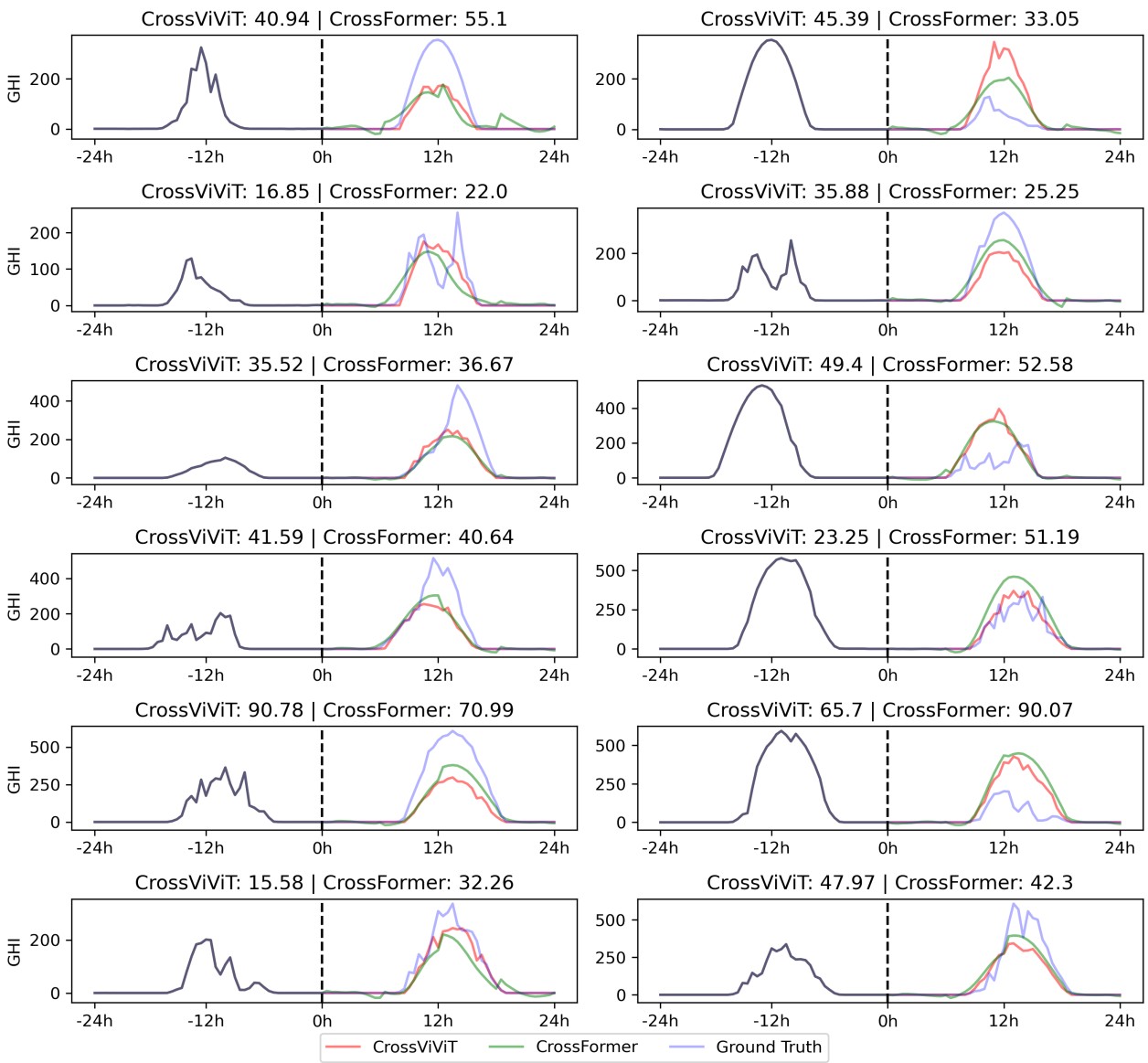

Figure 6: Visualizations of CrossViViT and CrossFormer are presented for a subset of 12 randomly selected days from the "Hard" split on the PAY station during the validation period spanning from 2017 to 2019.

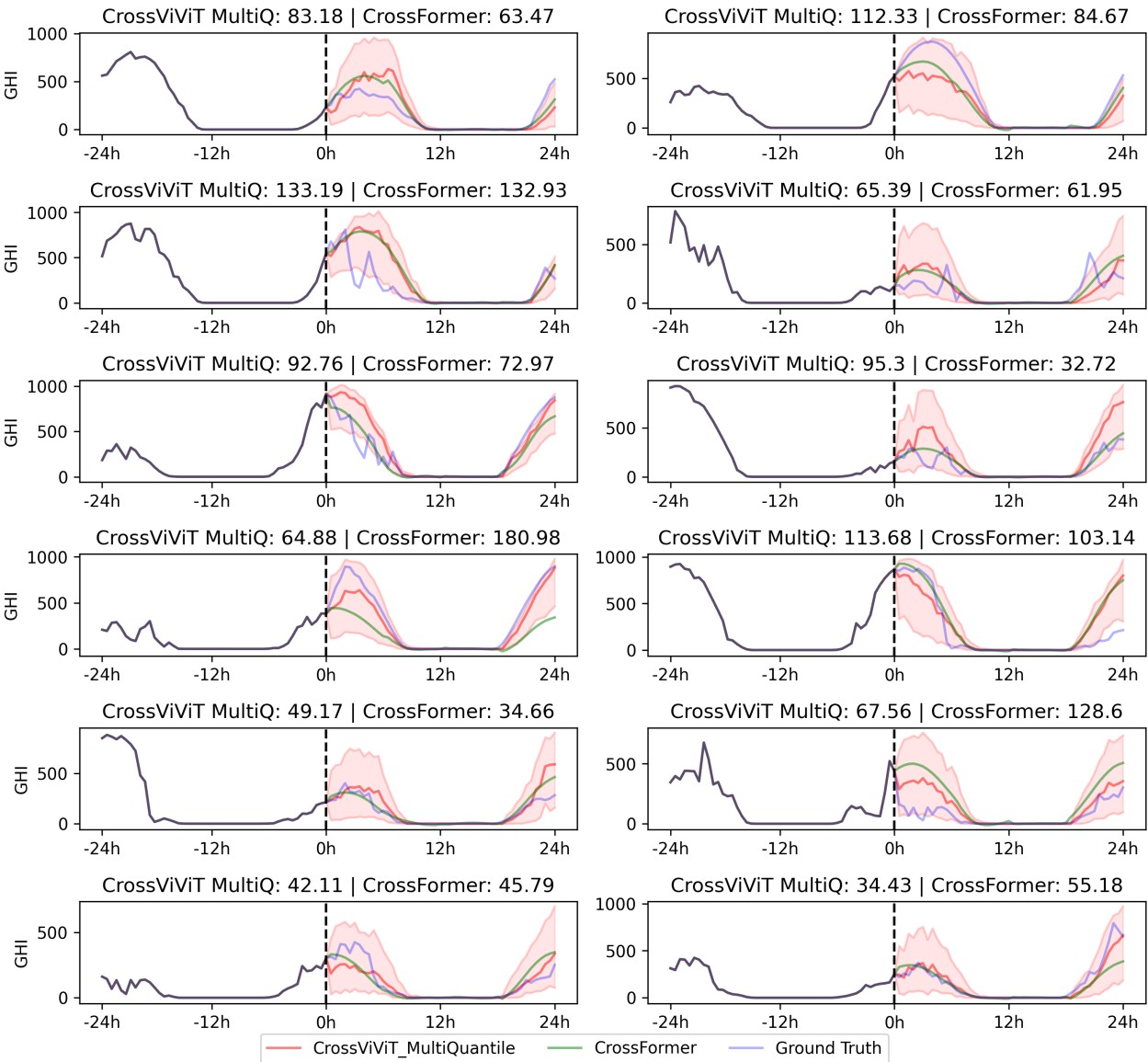

Figure 7: Visualizations of Multi-Quantile CrossViViT (median prediction and $[q_{0.02}, q_{0.98}]$ prediction interval) and CrossFormer predictions are presented for a subset of 12 randomly selected days from the "Hard" split on the PAY station during the validation period spanning from 2020 to 2022.

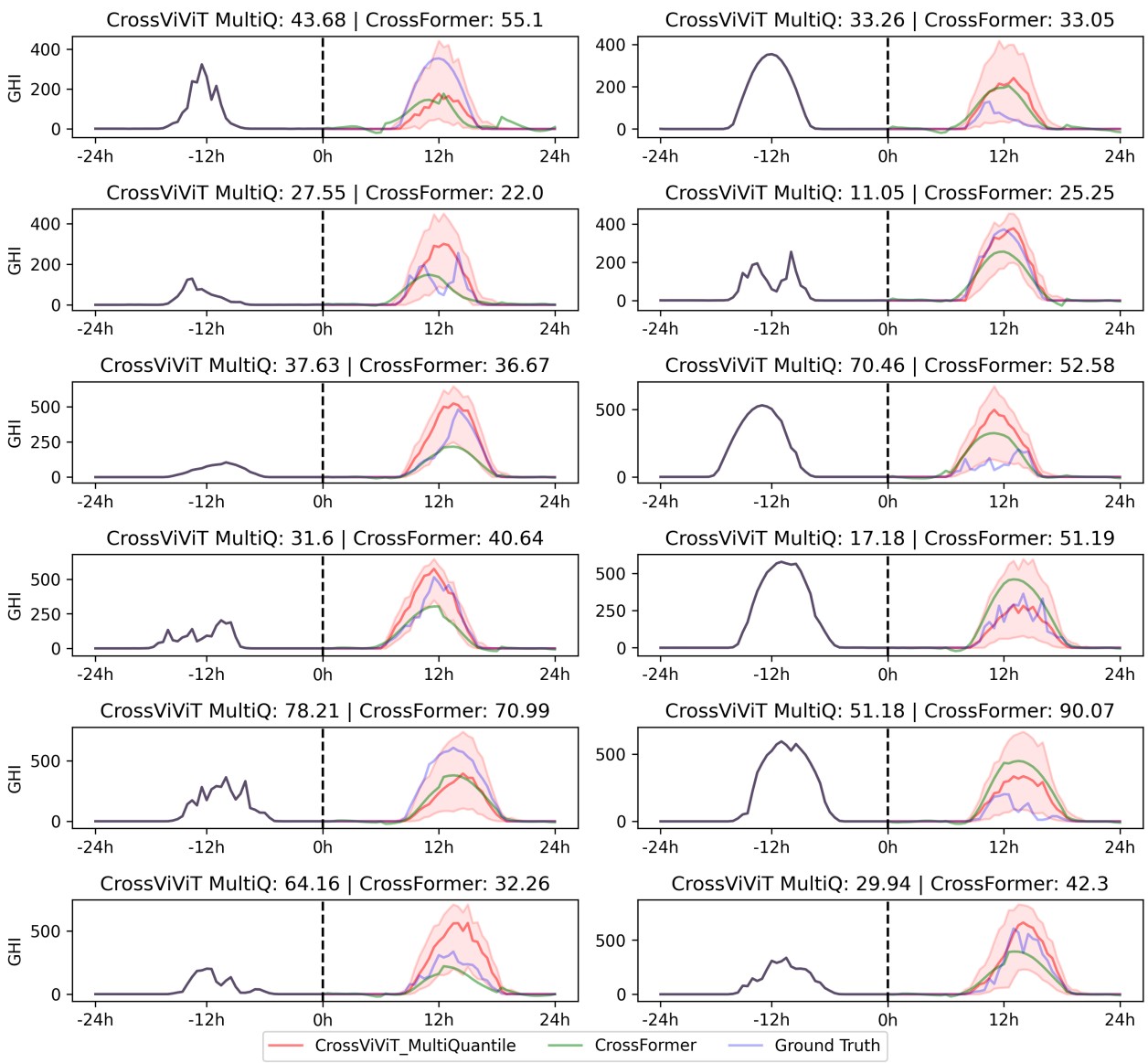

Figure 8: Visualizations of Multi-Quantile CrossViViT (median prediction and $[q_{0.02}, q_{0.98}]$ prediction interval) and CrossFormer predictions are presented for a subset of 12 randomly selected days from the "Hard" split on the PAY station during the validation period spanning from 2017 to 2019.

Table 6: Comparison of models' inference metrics and training times. We report the mean latency and standard deviation in milliseconds as well as Giga MACs and the training time per epoch. All these metrics were computed for a single NVIDIA RTX8000 GPU.

| Models | Latency (ms) | Giga MACs | Training time per epoch (s) |
|---|---|---|---|
| **ReFormer** (Kitaev et al., 2020) | $5.43 \pm 0.32$ | 1.66 | 387 |
| **Informer** (Zhou et al., 2021) | $8.63 \pm 0.30$ | 2.54 | 623 |
| **FiLM** (Zhou et al., 2022b) | $9.5 \pm 0.25$ | - | 445 |
| **PatchTST** (Nie et al., 2023) | $2.69 \pm 0.13$ | 0.57 | 106 |
| **LighTS** (Zhang et al., 2022) | $0.82 \pm 0.087$ | $< 0.01$ | 370 |
| **CrossFormer** (Zhang and Yan, 2023) | $17.89 \pm 0.43$ | 7.05 | 1300 |
| **FEDFormer** (Zhou et al., 2022a) | $66.41 \pm 0.63$ | 1.03 | 1134 |
| **DLinear** (Zeng et al., 2022) | $0.29 \pm 0.007$ | $< 0.01$ | 365 |
| **AutoFormer** (Wu et al., 2021) | $32.03 \pm 0.54$ | 2.92 | 1500 |
| **CrossViViT (145M)** | $65.03 \pm 0.43$ | 180.47 | 18000 |
| **Multi-Quantile CrossViViT (78.8M)** | $50.48 \pm 0.24$ | 100.45 | 18000 |

# C   Vision Transformers (ViT) and Video Vision Transformers (ViViT)

**ViT**   The Vision Transformer (ViT) model (Dosovitskiy et al., 2020) leverages self-attention mechanisms to efficiently process images, inspired by the popular transformer architecture (Bahdanau et al., 2015; Vaswani et al., 2017) notably based on the scaled dot product attention: given the query matrix $Q$, key matrix $K$, and value matrix $V$: $\text{Attention}(Q, K, V) = \text{softmax}\left(\frac{QK^T}{\sqrt{d_k}}\right)V$ where $d_k$ is the dimensionality of the keys. In the ViT model, the input image of dimensions $H \times W$ is divided into non-overlapping patches $x_i \in \mathbb{R}^{h \times w}$, which are linearly projected and transformed into $d$-dimensional vector tokens $z_i \in \mathbb{R}^d$ using a learned weight matrix $\mathbf{E}$ that applies 2D convolution. The sequence of patches is defined as $\mathbf{z} = [z_{\text{class}}, \mathbf{E}x_1, ..., \mathbf{E}x_P] + \mathbf{E}_{\text{pos}}$, where $z_{\text{class}}$ is an optional learned classification token representing the class label, and $\mathbf{E}_{\text{pos}} \in \mathbb{R}^{P \times d}$ is a 1D learned positional embedding that encodes position information.

To extract global features from the image, the embedded patches undergo $K$ transformer layers. Each transformer layer $k$ consists of Multi-Headed Self-Attention (MSA) (Vaswani et al., 2017), layer normalization (LN) (Ba et al., 2016), and MLP blocks with residual connections. The MLPs consist of two linear layers separated by the Gaussian Error Linear Unit (GELU) activation function (Hendrycks and Gimpel, 2016). The output of the final layer can be used for image classification, either by directly utilizing the classification token or by applying global average pooling to all tokens $\mathbf{z}^L$ if the classification token was not used initially.

**ViViT**   The Video Vision Transformer (Arnab et al., 2021) extends the ViT model to handle video classification tasks by incorporating both spatial and temporal dimensions within a transformer-like architecture. The authors propose multiple versions of the model, with a key consideration being how to embed video clips for attention computation.

Two approaches are presented: (1) *Uniform frame sampling* and (2) *Tubelet embedding*. The first method involves uniformly sampling $n_t$ frames from the video clip, applying the same ViT embedding method Dosovitskiy et al. (2020) to each 2D frame independently, and concatenating the resulting tokens into a single sequence. The second method extracts non-overlapping spatio-temporal "tubes" from the input volume and linearly projects them into $\mathbb{R}^d$, effectively extending ViT's embedding technique to 3D data, akin to performing a 3D convolution. Intuitively, the second method allows for the fusion of spatio-temporal information during tokenization, whereas the first method requires independent temporal fusion post-tokenization. Experimental results, however, show only slightly superior performance for the second method in specific scenarios, despite its significantly higher computational complexity (Arnab et al., 2021).

