# OpenReview forum: "Improving *day-ahead* Solar Irradiance Time Series Forecasting by Leveraging Spatio-Temporal Context"
_NeurIPS.cc/2023/Conference — NeurIPS 2023 poster_

### Official Review · Reviewer_WJ1J · 2023-06-11

**Soundness:** 3 good
**Presentation:** 2 fair
**Contribution:** 3 good
**Rating:** 6
**Confidence:** 4

**Summary:**

The work presents a multi-modal model, called CrossViViT, to perform day-ahead solar global horizontal irradiance predictions. In that, the model combines spatial information from satellites, i.e. RGB, IR and vapor channels, across Europe with time series information from six point-like stations, i.e. clear sky, pressure, direct normal irradiance, diffuse horizontal irradiance as well as a derived proxy global horizontal irradiance based on the Ineichen model. CrossViViT's performance has been compared again various other statistical and numerical models (Persistence and FFT) as well as other state-of-the-art deep learning approaches based on the transformer building blocks. The performance, measured in RMSE or MAE,

**Strengths:**

- Application case from natural sciences incl. challenging real-world problems of multi-modality and missing data
- Distinction between 'easy' and 'hard' prediction cases; this is commonly overlooked
- Open discussion of strength and limitations of CrossViViT in contrast to other method, in particular:
    * Showcasing that it does not win across the board
    * Improving the interesting 'hard' cases for domain applications

**Weaknesses:**

- The reviewer thinks that the evaluation in the manuscript could generally be improved with findings from other papers as follows:
    * Normalize that forecasting values into a stated range (e.g. [0-1]) or state the value ranges otherwise an RMSE/MAE improvement by certain value cannot be put into a frame of reference
    * Alternatively, consider reporting MAPE, mean average percentage, improvements instead that already includes that maximum range
    * By extension the plots in Fig. 4 can be considered somewhat misleading as they do not show the minimum value 0 or give a magnitude of improvement
- The authors dismiss the use of fourier-layers, e.g. AFNO, without clear substantiation
- Since cross-attention is such an integral part for mixing tokens from the spatial and temporal domains, it would be meaningful to show the equation in your manuscript beyond the reference
- It would be meaningful to extend the discussion of the findings towards the domain and/or real-world. What does it mean that you can achieve
- Non-adherence to the conference paper template, fonts in tables are too small, figures are outside of the text margins
- Color palette is difficult to read for color-blind people (red-green)
- The study is not reproducible as there is no code given; there is no indication that it will be released in case of acceptance
- Might be good to show the rough location, possibly with an arrow, for the TAM station on the map in Fig. 2
- To the reviewer's personal taste: change the title from a question to a typical title like "Solar Irradiance Time Series Forecasting with Spatio-Temporal Context"

**Questions:**

No direct question but an invitation to provide additional details or comments to the points raised in 'Weaknesses'

---

> ### Author Rebuttal · Authors · 2023-08-08
>
> > Normalize that forecasting values into a stated range (e.g. [0-1]) or state the value ranges otherwise an RMSE/MAE improvement by certain value cannot be put into a frame of reference
>
> We thank the reviewer for the suggestion. We think, however, that given that all the models are compared on the same setups and the same unobserved years for each station, the improvement of one model should be interpretable with respect to the others. However, if the reviewer means that normalizing would be helpful in order to compare the results across stations as well, we agree that it is a valid point. Yet, since the different stations have different patterns, we believe it is better to compare models for each station and not compare between stations, so that we keep in mind the order of magnitude of the values which can differ greatly from one station to another.
>
> > Alternatively, consider reporting MAPE, mean average percentage, improvements instead that already includes that maximum range
>
> Indeed, MAPE could be a good choice, but unfortunately it is problematic in our cases given that we are also considering nights through our sliding window training process, which involves ground truth values of 0, which cannot be handled by the MAPE by definition.
>
> > By extension the plots in Fig. 4 can be considered somewhat misleading as they do not show the minimum value 0 or give a magnitude of improvement
>
> The goal of the radar plots on Fig 4 is to compare the three models appearing on them, regarding each of the metrics we present. This, in our opinion, is easily doable using the plots. What would be the advantage of showing the minimum value 0?
>
> > The authors dismiss the use of fourier-layers, e.g. AFNO, without clear substantiation
>
> Does the reviewer mean mentioning fourier layers in related works, or using them within the architecture of CrossViViT? If it is the former, we did mention FNO, AFNO and GeoFNO in related works, as they are indeed very important works regarding weather prediction, PDE solving and video prediction in general. However, we feel that we wanted to build an architecture that only uses “basic” building blocks (including transformers, given the revolution they brought in our field) rather than already advanced architectures. There is of course a possibility to integrate many different existing methods to possibly improve the results, but we believe it is better to start without them at first. Furthermore, since we are not doing directly video prediction (although it would have been possible to predict the context as an auxiliary target, but we chose not to at first), we do not think fourier layers were necessarily first in line among all recent methods to improve our forecasting performance.
>
> > Since cross-attention is such an integral part for mixing tokens from the spatial and temporal domains, it would be meaningful to show the equation in your manuscript beyond the reference
>
> We thank the reviewer for this comment, we added the equation to the manuscript.
>
> > It would be meaningful to extend the discussion of the findings towards the domain and/or real-world. What does it mean that you can achieve
>
> We apply CrossViViT to a real-world problem of forecasting solar-irradiance which can be useful for mitigating climate change by encouraging the use of solar energy. We also mentioned in the introduction that CrossViViT can in principle be used to forecast any other physical variables.
>
> > Non-adherence to the conference paper template, fonts in tables are too small, figures are outside of the text margins
>
> We modified the paper to fit everything within the text margins, except for Figure 3 showing prediction visualizations, which would unfortunately be unreadable if keep it within the margins; if it is a problem, we can put some of the visualizations in the supplementary material.
>
> > Color palette is difficult to read for color-blind people (red-green)
>
> When making the plots, we tried to make sure that they were color-blind friendly as one of our co-authors is color-blind as well (and they made the plots). Can you give suggestions of colors we can use instead?
>
> > The study is not reproducible as there is no code given; there is no indication that it will be released in case of acceptance
>
> You can find the repository here: https://anonymous.4open.science/r/CrossViVit-57C2
>
> > Might be good to show the rough location, possibly with an arrow, for the TAM station on the map in Fig. 2
>
> Figure 2 was changed accordingly.
>
> > To the reviewer's personal taste: change the title from a question to a typical title like "Solar Irradiance Time Series Forecasting with Spatio-Temporal Context"
>
> The title was changed accordingly.

---

> > ### Comment · Reviewer_WJ1J · 2023-08-11
> >
> > > We thank the reviewer for the suggestion. We think, however, that given that all the models are compared on the same setups and the same unobserved years for each station, the improvement of one model should be interpretable with respect to the others. However, if the reviewer means that normalizing would be helpful in order to compare the results across stations as well, we agree that it is a valid point. Yet, since the different stations have different patterns, we believe it is better to compare models for each station and not compare between stations, so that we keep in mind the order of magnitude of the values which can differ greatly from one station to another.
> >
> > > Indeed, MAPE could be a good choice, but unfortunately it is problematic in our cases given that we are also considering nights through our sliding window training process, which involves ground truth values of 0, which cannot be handled by the MAPE by definition.
> >
> > It seems that reviewer and the authors have a different view on reporting numbers in regression tasks. For the reviewer, it is more meaningful to report a relative performance/improvement of a predictor rather than the absolute scale. This enables better judgment for individual stations, but also across the stations. This ties into the desire for a normalization/standardization of the scales as well as the the request to report the MAPE. While the later is challenging for small values, one can use typical numerical tricks like dividing by a very small epsilon.
> >
> > > The goal of the radar plots on Fig 4 is to compare the three models appearing on them, regarding each of the metrics we present. This, in our opinion, is easily doable using the plots. What would be the advantage of showing the minimum value 0?
> >
> > Not including a zero value for a bounded range allows to visually overly enhance improvements.
> >
> > > However, we feel that we wanted to build an architecture that only uses “basic” building blocks [...]. Furthermore, since we are not doing directly video prediction (although it would have been possible to predict the context as an auxiliary target, but we chose not to at first), we do not think fourier layers were necessarily first in line among all recent methods to improve our forecasting performance.
> >
> > The reviewer acknowledged the reasoning of the author for making a more focused study on basic building blocks. Nevertheless, Fourier layers have shown better predictive performance compared to standard blocks for networks such as ForecastNet. The authors may consider looking at them in the future.
> >
> > > We apply CrossViViT to a real-world problem of forecasting solar-irradiance which can be useful for mitigating climate change by encouraging the use of solar energy. We also mentioned in the introduction that CrossViViT can in principle be used to forecast any other physical variables.
> >
> > The reviewer would like to point out that the authors have only demonstrated a model with better predictive performance. Yet, what would it take to get this model in production? Can the possible effects for the climate be quantified? Is there a cost-use breakeven point? It would be meaningful to at least roughly reason about these or related aspects.
> >
> > > When making the plots, we tried to make sure that they were color-blind friendly as one of our co-authors is color-blind as well (and they made the plots). Can you give suggestions of colors we can use instead?
> >
> > There are several dedicated color palettes, e.g. to be found here: https://www.nceas.ucsb.edu/sites/default/files/2022-06/Colorblind%20Safe%20Color%20Schemes.pdf. To the personal taste of the reviewer Tol Muted works well, but others will probably also do.
> >
> > All other points: thanks you for incorporating them in the manuscript.

---

> > > ### Author Response · Authors · 2023-08-15
> > >
> > > >Indeed, MAPE could be a good choice, but unfortunately it is problematic in our cases given that we are also considering nights through our sliding window training process, which involves ground truth values of 0, which cannot be handled by the MAPE by definition.
> > > It seems that reviewer and the authors have a different view on reporting numbers in regression tasks. For the reviewer, it is more meaningful to report a relative performance/improvement of a predictor rather than the absolute scale. This enables better judgment for individual stations, but also across the stations. This ties into the desire for a normalization/standardization of the scales as well as the the request to report the MAPE. While the later is challenging for small values, one can use typical numerical tricks like dividing by a very small epsilon.
> > >
> > > We understand that a relative metric could be easier to showcase the improvement of a predictor. We therefore computed the MAPE for the predictions, using a small epsilon to replace 0 values for night time steps.
> > > Here, as a sample of the results, are the results for the day ahead prediction of our CrossViViT models and other models, on CAB test station and test years. As you can observe, CrossViViT models are still leading in performance. In the last version we will add the MAPE for all tests and cases.
> > > | Model          | MAPE (207) | MAPE Easy (120) | MAPE Hard (87) |
> > > |--------------------------|:----------:|:---------------:|:--------------:|
> > > | Persistence       |  0.54  |    0.34   |   0.82   |
> > > | Fourier_3        |  8.32  |    8.18   |   8.52   |
> > > | Fourier_4        |  5.18  |    5.44   |   4.82   |
> > > | Fourier_5        |   4   |    3.97   |   4.05   |
> > > | Clear Sky        |  0.72  |    0.48   |   1.05   |
> > > | Reformer         |   5.1  |    4.83   |   5.47   |
> > > | Informer         |   6.7  |    5.71   |   8.06   |
> > > | FiLM           |  7.43  |    7.8    |   6.91   |
> > > | PatchTST         |  2.44  |    2.35   |   2.57   |
> > > | LightTS         |   6.8  |    6.61   |   7.06   |
> > > | CrossFormer       |  3.24  |    2.93   |   3.68   |
> > > | FEDFormer        |  5.67  |    5.15   |   6.38   |
> > > | Dlinear         |  14.08  |   12.05   |   16.87   |
> > > | AutoFormer        |  13.04  |   12.63   |   13.6   |
> > > | CrossViViT        |  **0.45**  |    **0.31**   |   **0.64**   |
> > > | CrossViViT (No RoPE)   |  0.84  |    0.62   |   1.16   |
> > > | CrossViViT MultiQuantile |  0.62  |    0.4    |   0.93   |
> > > > Not including a zero value for a bounded range allows to visually overly enhance improvements.
> > >
> > > Following the last comment, We will also include MAPE in the radar plots in the last revision.
> > > >The reviewer acknowledged the reasoning of the author for making a more focused study on basic building blocks. Nevertheless, Fourier layers have shown better predictive performance compared to standard blocks for networks such as ForecastNet. The authors may consider looking at them in the future.
> > >
> > > We are definitely not underestimating the predictive power of Fourier layers, but solely thought they were not the most adapted for our first tentative. Yet, FourcastNet is indeed impressive and we’ll make sure to consider them for future efforts.
> > >
> > > >The reviewer would like to point out that the authors have only demonstrated a model with better predictive performance. Yet, what would it take to get this model in production? Can the possible effects for the climate be quantified? Is there a cost-use breakeven point? It would be meaningful to at least roughly reason about these or related aspects.
> > >
> > > We acknowledge the reviewer's concerns and we agree with the points made. Evaluating the usefulness of such models using metrics like MAE or RMSE certainly do not capture the nature of the forecasts, that's why we split the data into "Easy" cases and "Hard" cases and we agree that while it is a step in the right direction, it's not enough. The usefulness of such models can only be evaluated by looking at its downstream performance which in our cases would correspond to the amount of energy produced vs its carbon cost. This is a future work that we are envisioning by working on real stations and using our forecasts to guide the management of the energy grid. For the cost of the model, we will estimate its carbon footprint and add it to the camera-ready version. But it's important to keep in mind that such a model should in principle only be used once a day to produce the forecast and given the MACs that we reported, the potential savings in the generated energy's carbon footprint would outweigh that of CrossViViT.
> > >
> > > >There are several dedicated color palettes, e.g. to be found here: https://www.nceas.ucsb.edu/sites/default/files/2022-06/Colorblind%20Safe%20Color%20Schemes.pdf. To the personal taste of the reviewer Tol Muted works well, but others will probably also do.
> > >
> > > Following your recommendation, we will re-produce the results in the Tol Muted palette.
> > >
> > > >All other points: thanks you for incorporating them in the manuscript.
> > >
> > > Thanks!

---

### Official Review · Reviewer_eJfQ · 2023-06-25

**Soundness:** 3 good
**Presentation:** 3 good
**Contribution:** 3 good
**Rating:** 7
**Confidence:** 3

**Summary:**

The work presents a method to integrate information about cloud (using satellite images) with timeseries data related to Solar Irradiance to improve the solar irradiance forecasting.

**Strengths:**

Here some interesting aspects of the paper:
- the release of a new dataset containing both timeseries and satellite images for many years for several sites.
- an attempt to build a multimodal architecture based on transformer.
- usually the subdivision between sunny days and cloudy days is done in the forecasting works related to solar irradiance (e.g. pv production). Authors have proposed to subdivide the days in "hard" and "easy" based on the similarity between two consecutives days. I think this approach is interesting and help the fair assessment of the model.

**Weaknesses:**

The main weakness I see is that day-ahead use case is not explicitely evaluated.
I know that the sliding window is more general but an important real word use case is to have a real day-ahead prediction.
Authors could test their algorithm on day-ahead use case (extracting properly the sliding windows of interest, i.e. from 0:00 to 23:00).

Moreover, some details on the real input of the model seem missing.

**Questions:**

Fig.2 --> The TAM is outside the area of interest. This is described in the text but, since the table in fig. reports TAM but it is not present in the first satellite image, the authors should insert a comment in the caption.

Row 230: It is not true for TAM as wrote in row 266

Fig.3 and Fig.4 contain some discussions of the results but it would be better if these comments were moved in a proper section in the text.

It could be interesting to consider a day-ahead situation, in other words considering the sliding window starting at 0:00 and ending at 23.00 of day-1 and compute the next day. In this way the most common day-ahead use case is tested.

I think the satellite image are related to the same time of the time series. I didn't find this information in the text (or I missed it).

Some details are not clear to me.
1. the context image is used as a separated channel?
2. all data from all stations are used for the training?
3. What are effectively the inputs of the model, a list and a detailed encoding would be preferrable. I gave a look at the appendix, some details are present other not.
4. what are the computational time required for the training of the proposed approach?

An interesting future perspective could be to incorporate the forecasting of atmosphere's state (images obtained by a Weather Prediction service provider).

Row 51
Sound prediction ?

**Limitations:**

Some limitations have been identified and discussed by authors.

---

> ### Author Rebuttal · Authors · 2023-08-08
>
> > The main weakness I see is that day-ahead use case is not explicitely evaluated. I know that the sliding window is more general but an important real word use case is to have a real day-ahead prediction. Authors could test their algorithm on day-ahead use case (extracting properly the sliding windows of interest, i.e. from 0:00 to 23:00).
>
> Thank you for the suggestion, we did the test for day-ahead use case, practically selecting only the 00:00 to 23:00 windows when computing the metrics. The resulting table can be found in the rebuttal PDF.
>
> As the reviewer can see, it only slightly decreases the absolute performance of all models but comparatively, CrossViViT still outperforms the baselines. While we agree that this demonstrates the performance in an operational setting, we believe it would fit better in the appendix rather than the main text.
>
> > Moreover, some details on the real input of the model seem missing.
>
> We are not sure we understand what the reviewer means. What type of details are missing?
>
> >**Fig.2 --> The TAM is outside the area of interest. This is described in the text but, since the table in fig. reports TAM but it is not present in the first satellite image, the authors should insert a comment in the caption.
>
> We thank the reviewer for this suggestion and agree with the premise. To clarify the position of TAM (slightly) outside of the area of interest, we increased the area covered by Figure 2 top left figure, including TAM, but highlighting the area we are considering in red. We also inserted a comment in the caption.
>
> > Row 230: It is not true for TAM as wrote in row 266
>
> This was corrected.
>
> > Fig.3 and Fig.4 contain some discussions of the results but it would be better if these comments were moved in a proper section in the text.
>
> Most comments are in the discussion part of the paper, yet we thought it would make for a better reading experience to also put some significant comments directly in the caption, with the figure of interest close by, rather than looking for the figure corresponding to the comment. It is in our opinion not uncommon to discuss results in captions.
>
> > I think the satellite image are related to the same time of the time series. I didn't find this information in the text (or I missed it).
>
> Yes, the time series and the satellite images are aligned temporally. We clarified that in the text.
>
> > the context image is used as a separated channel?
>
> What we call the spatial context is the ensemble of satellite data, which include multiple video channels, as described in the satellite data section. There are 11 channels in total.
>
> > all data from all stations are used for the training?
>
> As mentioned in the paper: 3 stations (IZA, CNR, PAL) are used for training, 1 for validation (PAY) and the two remaining for testing (TAM and CAB). It allows us to evaluate the spatial generalization capabilities of the model.
>
> > What are effectively the inputs of the model, a list and a detailed encoding would be preferrable. I gave a look at the appendix, some details are present other not.
>
> We presented all inputs in the method section and the type of data in the satellite data and time series section. Other details are indeed in the appendix. If you think more details are missing, we will be happy to add them in the appendix.
>
> > what are the computational time required for the training of the proposed approach?
>
> The table below highlights all the inference and training metrics for all models (Note that all the times presented there are for **one GPU**):
>
> |Model|Mean Latency (ms)|STDev. Latency (ms)|Giga MACs|Training time per epoch (s)|
> |---|---|---|---|---|
> |Reformer|5.43|0.32|1.66|387|
> |Informer|8.63|0.30|2.54|623|
> |FiLM|9.5|0.25||445|
> |PatchTST|2.69|0.13|0.57|106|
> |LightTS|0.82|0.087|0.0004|370|
> |CrossFormer|17.89|0.43|7.05|1300|
> |FEDFormer|66.41|0.63|1.03|1134|
> |DLinear|0.29|0.007|0.00004|365|
> |AutoFormer|32.03|0.54|2.92|1500|
> |CrossViViT|65.03|0.43|180.47|18000|
> |CrossViViT MultiQuantile|50.48|0.24|100.45|18000|
>
> *Latency: The time it takes for the model to process one instance (batch size=1)*
> *MAC: Number of multiply-accumulate operations. A multiply-accumulate operation corresponds to the operation a+(b*c) which counts as one operation.
>
> We don't report MACs for FiLM since it's based on S4 model and the library we used doesn't support it.
>
> > An interesting future perspective could be to incorporate the forecasting of atmosphere's state (images obtained by a Weather Prediction service provider).
>
> It is indeed in our plan for future work! Thanks for the suggestion.
>
> > Row 51 Sound prediction ?
>
> By sound prediction we meant correct prediction; We replaced it in the text to avoid any confusion.

---

> > ### Comment · Reviewer_eJfQ · 2023-08-12
> >
> > > As the reviewer can see, it only slightly decreases the absolute performance of all models but comparatively, CrossViViT still outperforms the baselines. While we agree that this demonstrates the performance in an operational setting, we believe it would fit better in the appendix rather than the main text.
> >
> > Ok
> >
> > > We are not sure we understand what the reviewer means. What type of details are missing?
> >
> > Some details that I have asked for in the other questions.
> >
> > > We presented all inputs in the method section and the type of data in the satellite data and time series section. Other details are indeed in the appendix. If you think more details are missing, we will be happy to add them in the appendix.
> >
> > I think the appendix contains a good amount of information.
> >
> > >The table below highlights all the inference and training metrics for all models (Note that all the times presented there are for one GPU)
> >
> > Ok
> >
> > The authors have answered my questions and solved my doubts. I increased my rating accordingly.

---

> > > ### Author Response · Authors · 2023-08-14
> > > **Follow up answer**
> > >
> > > Thank you for your feedback. Following your questions as well as those from all other reviewers, we added many clarifications and details to the manuscript, hopefully making the last version of the paper as clear as possible.

---

### Official Review · Reviewer_zf5t · 2023-06-29

**Soundness:** 3 good
**Presentation:** 3 good
**Contribution:** 3 good
**Rating:** 6
**Confidence:** 4

**Summary:**

This submission presents a multimodal model for next-day solar irradiance prediction. They use time series of past irradiance and satellite image to predict irradiance 24h in advance. The model consists of one transformer branch for each modality and a shared (cross-modal) transformer. Their method can be used to predict uncertainty as well. They display improvements over the SOTA. Finally, they released  in open access a dataset acquired with 6 stations across 15 years.

**Strengths:**

- The problem is interesting, difficult, useful, and not a lot of work has been done with machine learning on the subject

- The model architecture is reasonable

- the authors compare their method to many baselines and competing methods

- The authors provide a large-scale (at least temporally) dataset in open access

**Weaknesses:**

- Some details are missing, making it hard to understand how the method works precisely.

- The uncertainty prediction with quantile lacks a proper evaluation, related work, and comparison baselines.

**Questions:**

Q1) the cross-attention module needs to better explained. There are TxNp video tokens and T temporal tokens; how are they mixed? The authors use "learned positional encoding" for these tokens, but it is not explained how. Is the absolute spatiotemporal position encoded? The different sampling rates of eumetsat (5min) and bsrn (1h), yet they have the same number of observations T?

Q2) What is the influence of ROPE compared to a more standard positional encoding? Since the authors make it an integral part of their method, its effect should be quantified in an ablation study

Q3) we allow the model to mask a portion of the past time-series -> Why would the model want to do that? That can only decrease the train performance

S1) The title shouldn't be a question, especially such a hyper-precise and niche one! It should be: Improved Day-Ahead Solar Irradiance Time Series Forecasting by Leveraging Spatio-Temporal Context or something in that vein

**Limitations:**

not provided

---

> ### Author Rebuttal · Authors · 2023-08-08
>
> We thank the reviewer for the thoughtful feedback and suggestions to improve the paper. We respond to the reviewer's comments below:
>
> > Some details are missing, making it hard to understand how the method works precisely.
>
> We are not sure we understand what the reviewer means. What type of details are missing?
>
> > The uncertainty prediction with quantile lacks a proper evaluation, related work, and comparison baselines.
>
> The Multi Quantile version serves as a supplementary version to offer a way to extract the uncertainty attached to the prediction, which can be implemented in most deep learning architectures, but is not the main contribution of the work, so we had to keep the results and comparison light in this regard. However, we do evaluate its median prediction (even though it is not meant as the definitive prediction) and its ability to include the observed true values. Do you have suggestions for more things we can include?
>
> > Q1) the cross-attention module needs to better explained. There are TxNp video tokens and T temporal tokens; how are they mixed?
>
> We first mix tokens spatially, that is we have no regard for time initially which is implemented by simply reshaping the time dimension into the batch size. This leaves for each time-step mixing $N$ context tokens with one time-series token which is done using cross-attention and results in one token. We believe this was clearly stated in the methods section and illustrated in Figure 1, but if not, let us know how we can make it clearer.
>
> > The authors use "learned positional encoding" for these tokens, but it is not explained how. Is the absolute spatiotemporal position encoded?
>
> The mixing is done first spatially by considering each time point separately.  We first mix the tokens spatially using cross-attention through Rotary Positional Encoding (RoPE) and then the resulting tokens are concatenated in a temporal sequence which is processed by a transformer that adds a learned positional encoding. This is similar to how Video ViT works, as described in the appendix.
>
> > The different sampling rates of eumetsat (5min) and bsrn (1h), yet they have the same number of observations T?
>
> The original sampling rates of EUMETSAT is 5 min and BSRN is 1 minute, we down-sample them to 30min each, so that both context and time-series have a sampling rate of 30 minutes.
>
> > Q2) What is the influence of ROPE compared to a more standard positional encoding? Since the authors make it an integral part of their method, its effect should be quantified in an ablation study
>
> We thank the reviewer for this observation. We did the ablation and will include the results in the paper. The findings summarized in the table below (which extends Table 1) suggest that RoPE does help improve the performance.
>
> On CAB (2020-2022):
>
> | Model                   | MAE       | RMSE      | MAE Easy  | RMSE Easy | MAE Hard  | RMSE Hard  |
> |-------------------------|-----------|-----------|-----------|-----------|-----------|------------|
> | CrossViViT              | **50.35** | **99.18** | **47.04** | **89.60** | **55.30** | **112.00** |
> | CrossViViT without RoPE | 51.11     | 103.66    | **47.31** | 95.13     | 56.84     | 115.31     |
>
> On TAM (2017-2019):
>
> | Model                   | MAE       | RMSE      | MAE Easy  | RMSE Easy | MAE Hard  | RMSE Hard  |
> |-------------------------|-----------|-----------|-----------|-----------|-----------|------------|
> | CrossViViT              | **49.46** | **94.96** | **44.01** | **79.91** | 97.40     | **179.30** |
> | CrossViViT without RoPE | 109.28    | 196.44    | 111.33    | 197.63    | **91.29** | 185.61     |
>
> > Q3) we allow the model to mask a portion of the past time-series -> Why would the model want to do that? That can only decrease the train performance
>
> The idea behind the possibility of masking the past time series was to encourage the model to use the spatial context, rather than relying too much on the past time series, which is the first natural thing to do for the model. In particular, we wanted to prevent the model from simply repeating the past, making persistence-like predictions.
> Interestingly, we realized that even when masking entirely the past time series, the predictions were quite good, therefore showing that the model was indeed using the spatial context. In practice, however, the performance was still slightly better when keeping the time series masking to 0, so we kept it this way (while mentioning the possibility in the text and figures).
>
> > S1) The title shouldn't be a question, especially such a hyper-precise and niche one! It should be: Improved Day-Ahead Solar Irradiance Time Series Forecasting by Leveraging Spatio-Temporal Context or something in that vein.
>
> We will change the title accordingly.
>
> > Limitations: not provided
>
> We did mention a few limitations! Yet, it was rather limited, so we increased this part a little since the review.

---

> > ### Comment · Reviewer_zf5t · 2023-08-11
> > **Follow up questions.**
> >
> > > What type of details are missing?
> >
> > The ones that lead to the questions above.
> >
> > > The Multi Quantile version[...] is not the main contribution of the work, so we had to keep the results and comparison light in this regard.
> >
> > Multi Quantile is either a contribution, or it is not. Since you put it second in the list of contributions, it appears to be one. It would help if you placed this work with respect to the relevant literature and compared its performance with relevant approaches. If no existing work applies, explain why.
> >
> > > we do evaluate its median prediction [...] and its ability to include the observed true values.
> >
> > None of that measures the uncertainty estimation quality, the module's main goal.
> >
> > > Do you have suggestions for more things we can include?
> >
> > Here is a non-exhaustive list of relevant work. Explain the relation of your work with them, and look at how they evaluate the quality of the uncertainty estimation.
> >
> > Lakshminarayanan etal . Simple and scalable predictive uncertainty estimation using deep ensembles. Neurips, 2017.
> >
> > Turkoglu etal  FiLM-ensemble: Probabilistic deep learning via featurewise linear modulation. Neurips, 2022
> >
> > Gal etal  Dropout as a Bayesian approximation: Representing model uncertainty in deep learning. ICLR 2016
> >
> > > CrossViViT without RoPE
> >
> > Reading the other rebuttal, I see that you replaced RoPE it with "a learned PE". Can you give details about this baseline? How do you learn the positional encoding, is it a function? Why not use a standard Fourrier-based encoding?
> >
> > The other explanation does clarify my understanding of token mixing. The authors should make sure to improve the clarity of the text to reflect these non-trivial details.
> >
> > I would be inclined to increase my rating if the RoPE experiment turned out to be valid, and if the authors can correctly situate and evaluate their uncertainty prediction module (or explain convincingly why they cannot).

---

> > > ### Author Response · Authors · 2023-08-14
> > > **Follow up answers**
> > >
> > > > What type of details are missing?
> > >
> > > >   The ones that lead to the questions above.
> > >
> > > We will add details regarding the tokens mixing, as described in the rebuttal, as well as precisions regarding RoPE and many other details that were asked by the other reviewers. We hope that will result in a final version with significantly improved clarity with respect to the original draft.
> > >
> > > > The Multi Quantile version[...] is not the main contribution of the work, so we had to keep the results and comparison light in this regard.
> > >
> > > > Multi Quantile is either a contribution, or it is not. Since you put it second in the list of contributions, it appears to be one. It would help if you placed this work with respect to the relevant literature and compared its performance with relevant approaches. If no existing work applies, explain why.
> > >
> > > We agree with the reviewer. We therefore added a literature review on uncertainty extraction in general, and related to regression and forecasting tasks in the manuscript (unfortunately due to space issues, and because we cannot update the rebuttal pdf, we cannot include it entirely but you will be able to observe it in the next revision) Only very few approaches actually exist to attach prediction intervals to regression estimates, as it will be mentioned in the additional literature review. Ensemble and bootstrap methods would really not be the most efficient here as our model is still a bit heavier than model typically used with such methods (regression trees etc), and the approach suggested by Metnet (Sønderby, Casper Kaae, et al. "Metnet: A neural weather model for precipitation forecasting." arXiv preprint arXiv:2003.12140 (2020).), meaning separating the prediction into multiple value bins, and predicting the probability of each of the bins for each time step, might work, but would be in our opinion less expressive, as one map head would predict multiple bins instead of having multiple specialized heads. It is in a way a generalization of the same concept.
> > >
> > > >  We do evaluate its median prediction [...] and its ability to include the observed true values.
> > > >  None of that measures the uncertainty estimation quality, the module's main goal.
> > >
> > > When estimating prediction intervals, it is common practice to evaluate the quality of the estimation by examining the fraction of test points that fall inside the corresponding prediction intervals, which is precisely what we do. Note that most works, including the ones you mention in your next comment, tackle the evaluation of uncertainty regarding classification tasks (meaning taking into account the uncertainty of probabilistic outputs when converting them into classes) and only a handful tackle its regression counterpart. Note also that the evaluation of such prediction intervals attached to regression estimates is surprisingly challenging and a topic of research on itself, as pointed out by Sluijterman et al, 2023 (« How to Evaluate Uncertainty Estimates in Machine Learning for Regression? »), in a very recent work.  Sluijterman et al acknowledges that is one of the most common ways of doing so, yet advocates for simulation-based approaches. We will definitely explore these new ways of evaluation in the future, but for the time being for time concerns, we shall keep our evaluation scheme as it is if it is ok with the reviewer.
> > >
> > > > Do you have suggestions for more things we can include?
> > >
> > > >Here is a non-exhaustive list of relevant work[...]
> > >
> > > Thanks a lot for these suggestions. We included them in the new related works paragraph regarding uncertainty extraction, along with many other references closely related to our problem.
> > >
> > > > CrossViViT without RoPE
> > >
> > > > Reading the other rebuttal, I see that you replaced RoPE it with "a learned PE". Can you give details about this baseline? [...]
> > > I would be inclined to increase my rating if the RoPE experiment turned out to be valid, and if the authors can correctly situate and evaluate their uncertainty prediction module (or explain convincingly why they cannot).
> > >
> > > Indeed, RoPE is ablated against a standard learnable positional encoding as was done in the Video VIT and VIT papers. The learnable positional encoding is a learnable parameter $\mathbf{p}\in\mathbb{R}^{N\times d}$ where $N$ is the number of tokens in the sequence and $d$, the embedding dimension. We chose the learnable positional encoding scheme instead of the fourier-based one because the former is a standard choice and can be more felxible (if correctly learned) than a static embedding.

---

### Official Review · Reviewer_1GwN · 2023-07-05

**Soundness:** 3 good
**Presentation:** 3 good
**Contribution:** 3 good
**Rating:** 7
**Confidence:** 4

**Summary:**

The paper proposes a transformer-based day-ahead forecasting model for solar irradiance at a ground station.
The model ingests previous irradiances and contextual (image-sequence) information with a temporal and vision transformer.
A cross-former merges the tokens, and a temporal transformer decoder estimates irradiances in a 24-hour window.

An optional multi-quantile output head also allows the model to estimate uncertainty by forecasting quantiles.
This multi-quantile loss produces predictions with slightly lower accuracy, which may be justifiable at the benefit of uncertainty quantification.

Further considerations involve rotary positional encodings for the context image information, which is motivated by the images centered on the measurement station.

Overall, the paper presents a combination of state-of-the-art methods (ViVIT, transformers) with problem-specific ideas (ROPE, Quantile regression) crafted towards a suitable application (irradiance forecasting).

**Strengths:**

* important application (irradiance forecast) addressed with state-of-the-art machine learning (temporal + vision) and architecture crafted towards the forecasting application.
* uncertainty quantification with a loss function inspired by quantile regression (Koenker & Hallock, 2001)
* separate evaluation in easy and hard cases. Comparison to reasonable comparison methods and baselines
* evaluations between different stations to test out-of-domain generalization

**Weaknesses:**

* some design decisions are not justified or unclear (see questions: learned positional encoding).
* fast/sloppy preparation of some parts manuscript:
  * errors/typos in domain-specific equations: GHI = DNI + DHI x cos(z) <- I believe the x cos(z) should be with the DNI (direct normal irradiance) and not the DHI to account for the sun angle.
  * equation 7: shouldnt it be \hat{y}_\alpha, as there is a prediction \hat{y} for each quantile \alpha?
  * style and references are not consistent and some are not retrievable (ArXiv.org vs ArXiv, abs/2010.08895. Or “Rothfuss, H. (2015); Data access at eumetsat.” has no meaning and can not be retrieved); is “348 [BSRN] BSRN. Baseline surface radiation network.” from A Driemel · 2018 <- Cited by 249,  Baseline Surface Radiation Network (BSRN) ?

**Questions:**

* Why did the authors use a learned positional encoding for the temporal transformer rather than a regular periodic function?
* Why does Multi-Quantile CrossVivit have fewer parameters (almost half) than CrossVivit? Following my intuition, it should have more, as it has more MLP heads. Were the hyperparameters (i.e., number of layers, etc) different?
* According to the DNI/DHI Equation: Can the authors verify this is just a typo not implemented in the data generation?
* How applicable is the method compared to related work? In particular with respect to the runtime and inference time. Can the authors provide some numbers on the runtime compared to other approaches

**Limitations:**

A limitations section is included in the conclusion. However, these limitations are rather phrased toward future work and, for instance, describe a lack of data that is hard to address.
Other potential limitations like computational runtime, which is an important factor towards the applicability of this method, are not discussed. I feel this should be somewhat considered given that the proposed method is crafted for a particular application field.

---

> ### Author Rebuttal · Authors · 2023-08-08
>
> We thank the reviewer for the thoughtful feedback and suggestions to improve the paper. We respond to the reviewer's comments below:
>
> > errors/typos in domain-specific equations: GHI = DNI + DHI x cos(z) <- I believe the x cos(z) should be with the DNI (direct normal irradiance) and not the DHI to account for the sun angle.
>
> It was a typo indeed, we double-checked the data-generation process and it contained the right formula.
>
> > equation 7: shouldnt it be \hat{y}_\alpha, as there is a prediction \hat{y} for each quantile \alpha?
>
> There is indeed a different prediction for each quantile, so we changed the notation regarding the MultiQuantile loss, just under the equation 7 (yet not for equation 7 itself which simply defines a single quantile loss).
>
> > style and references are not consistent and some are not retrievable (ArXiv.org vs ArXiv, abs/2010.08895. Or “Rothfuss, H. (2015); Data access at eumetsat.” has no meaning and can not be retrieved); is “348 [BSRN] BSRN. Baseline surface radiation network.” from A Driemel · 2018 <- Cited by 249, Baseline Surface Radiation Network (BSRN) ?
>
> The three above points were corrected.
>
> > Why did the authors use a learned positional encoding for the temporal transformer rather than a regular periodic function?
>
> This is a design choice, motivated by the current literature on transformers. In particular, it was used in the architecture that partly gave us inspiration, the Video VIT (https://arxiv.org/pdf/2103.15691.pdf), as mentioned in the paper. It is therefore natural that we employ the same positional encoding here, as it appears to offer more expressivity and function the best.
>
> > Why does Multi-Quantile CrossVivit have fewer parameters (almost half) than CrossVivit? Following my intuition, it should have more, as it has more MLP heads. Were the hyperparameters (i.e., number of layers, etc) different?
>
> We tried to train the Multi quantile version with the same number of parameters, and the performance was simply a bit lower. The version shown in the paper, with different hyperparameters indeed (using smaller number of layers and smaller dimensionality as shown in the appendix), seemed to work better. Please note that in the case of this version, the goal is not to offer the best median prediction, but rather the best confidence interval, meaning with a highest probability of including the observed values, which was offered by the smaller version. For completeness however, we added the results of the larger model, with the same hyperparameters as the normal CrossViViT, in the tables. This version has 145.5M parameters (a little bit more than the normal CrossViViT, since, as pointed out, it has more MLP heads).
>
> > How applicable is the method compared to related work? In particular with respect to the runtime and inference time. Can the authors provide some numbers on the runtime compared to other approaches
>
> Below, we present a table with different inference and training metrics:
>
> | Model                    | Mean Latency (ms) | STDev. Latency (ms) | Giga MACs | Training time per epoch (s) |
> |--------------------------|-------------------|---------------------|-----------|-----------------------------|
> | Reformer                 | 5.43              | 0.32                | 1.66      | 387                         |
> | Informer                 | 8.63              | 0.30                | 2.54      | 623                         |
> | FiLM                     | 9.5               | 0.25                |           | 445                         |
> | PatchTST                 | 2.69              | 0.13                | 0.57      | 106                         |
> | LightTS                  | 0.82              | 0.087               | 0.0004    | 370                         |
> | CrossFormer              | 17.89             | 0.43                | 7.05      | 1300                        |
> | FEDFormer                | 66.41             | 0.63                | 1.03      | 1134                        |
> | DLinear                  | 0.29              | 0.007               | 0.00004   | 365                         |
> | AutoFormer               | 32.03             | 0.54                | 2.92      | 1500                        |
> | CrossViViT               | 65.03             | 0.43                | 180.47    | 18000                       |
> | CrossViViT MultiQuantile | 50.48             | 0.24                | 100.45    | 18000                       |
>
> *Latency: The time it takes for the model to process one instance (batch size=1)*
> *MAC: Number of multiply-accumulate operations. A multiply-accumulate operation corresponds to the operation a+(b*c) which counts as one operation.
> As it can be observed on the table, the lightest models are indeed very fast and theoretically much faster than Ours, CrossViViT. However, all models have an inference time that lies within 1 second, and in practice it therefore does not make such a difference.
> We would like to note within the context of a solar installation, day-ahead forecasting would be done once a day for each station, so as long as the inference time is not unreasonably high, which is the case here, it seems acceptable to sacrifice inference time for better performance, given that time-series models are less compute-hungry in comparison.
> Given the previous comments, we suggest not including it in the main text, and instead in the appendix, to avoid making the tables too heavy.
>
> > Limitations
>
> A small note on this was added to the conclusion.

---

> > ### Comment · Reviewer_1GwN · 2023-08-13
> > **Thank you and follow-up on limitation discussion**
> >
> > Thank you for providing detailed answers to my questions.
> >
> > There are no major disagreements left, and I would be glad to see the paper accepted at the conference as it addresses an interesting problem field with state-of-the-art models and is technically solid.
> >
> > As remark on the limitations discussion:
> > As the paper appears sound, a more detailed discussion of existing limitations would be appreciated and highly beneficial to guide future research. A lack of description of limitations in the initial paper was also mentioned by reviewer zf5t and adding "small note" in the conclusion (reply to my review) or increasing the "part a little since the review." (response to reviewer zf5t) is not satisfactory.
> >
> > While I appreciate the additional runtime results, they also show that the training time is > x30 longer, and the latency of transformer models is still 2x to 5x longer than classical models. This can be openly discussed as not a limitation of this work but rather of transformers in general. Hence, discussing this limitation openly would encourage future work to develop more efficient and still accurate models. This also touches on fairness questions with regard to access to computational resources.
> >
> > I trust that the authors integrate a thorough discussion on limitations in the final camera-ready version.

---

> > > ### Author Response · Authors · 2023-08-14
> > > **Follow up answers**
> > >
> > > Thank you very much for your feedback and suggestions.
> > > We will make sure to improve the limitations section with a longer discussion, including a part on the training and inference times of transformers and our model in particular, and relating it to the current challenges regarding the access to compute power. It is indeed a very significant subject to tackle.

---

### Author Rebuttal · Authors · 2023-08-08

We thank the reviewers for their efforts and reviews of high quality. We are happy that they appreciated our contributions to the forecasting literature, including our architecture and the “easy and hard” cases evaluation.

We carefully considered your suggestions, and believed that it ultimately made our paper better; we therefore thank you again for it. We answered each reviewer separately, point by point, using the discussion tool.

In a nutshell, these are the main points and additions made to the original manuscript:

- Ablated the RoPE Positional Encoding (replacing it by a learned PE) to measure its impact on the architecture:
On CAB (2020-2022):

| Model                   | MAE       | RMSE      | MAE Easy  | RMSE Easy | MAE Hard  | RMSE Hard  |
|-------------------------|-----------|-----------|-----------|-----------|-----------|------------|
| CrossViViT              | **50.35** | **99.18** | **47.04** | **89.60** | **55.30** | **112.00** |
| CrossViViT without RoPE | 51.11     | 103.66    | **47.31** | 95.13     | 56.84     | 115.31     |

On TAM (2017-2019):

| Model                   | MAE       | RMSE      | MAE Easy  | RMSE Easy | MAE Hard  | RMSE Hard  |
|-------------------------|-----------|-----------|-----------|-----------|-----------|------------|
| CrossViViT              | **49.46** | **94.96** | **44.01** | **79.91** | 97.40     | **179.30** |
| CrossViViT without RoPE | 109.28    | 196.44    | 111.33    | 197.63    | **91.29** | 185.61     |
- Added results for a larger Multi-Quantile version of the model, matching the size of CrossViViT
On CAB (2020-2022):

| Model                             | Parameters | MAE   | $p_t$  | MAE Easy | $p_t$  | MAE Hard | $p_t$  |
|-----------------------------------|------------|-------|------|----------|------|----------|------|
| Multi-Quantile CrossViViT (small) | 78.8M      | 61.8  | 0.91 | 57.03    | 0.93 | 68.94    | 0.9  |
| Multi-Quantile CrossViViT (large) | 145.5M     | 74.26 | 0.89 | 68.83    | 0.91 | 82.39    | 0.87 |

On TAM (2017-2019):

| Model                             | Parameters | MAE   | $p_t$  | MAE Easy | $p_t$  | MAE Hard | $p_t$  |
|-----------------------------------|------------|-------|------|----------|------|----------|------|
| Multi-Quantile CrossViViT (small) | 78.8M      | 81.2  | 0.71 | 78.93    | 0.70 | 101.18    | 0.75  |
| Multi-Quantile CrossViViT (large) | 145.5M     | 79.73 | 0.76 | 76.08    | 0.76 | 111.74    | 0.75 |
- Added a table showing inference times of all models considered including our own, as well as training times

|Model|Mean Latency (ms)|STDev. Latency (ms)|Giga MACs|Training time per epoch (s)|
|---|---|---|---|---|
|Reformer|5.43|0.32|1.66|387|
|Informer|8.63|0.30|2.54|623|
|FiLM|9.5|0.25||445|
|PatchTST|2.69|0.13|0.57|106|
|LightTS|0.82|0.087|0.0004|370|
|CrossFormer|17.89|0.43|7.05|1300|
|FEDFormer|66.41|0.63|1.03|1134|
|DLinear|0.29|0.007|0.00004|365|
|AutoFormer|32.03|0.54|2.92|1500|
|CrossViViT|65.03|0.43|180.47|18000|
|CrossViViT MultiQuantile|50.48|0.24|100.45|18000|
- Added CrossViViT results for day-ahead only cases. Due to the character limit, we refer the reviewers to the attached rebuttal PDF that contains results for CAB (2020-2022) and TAM (2017-2019).
- Corrected the text, added some details and made a few things clearer about the method in general.
- Changed the name of the paper: *Improved Day-Ahead Solar Irradiance Time Series Forecasting by Leveraging Spatio-Temporal Context*.
- We added the anonymous repository to our code: https://anonymous.4open.science/r/CrossViVit-57C2/README.md

---

### Decision · Program_Chairs · 2023-09-21

**Decision:**

Accept (poster)

**Comment:**

This paper proposes a transformer-based model which leverages satellite data to provide the spatio-temporal context for solar irradiance time series forecasting. It also makes available a new multi-modal dataset that can benefit subsequent research by others.

We all agree that the application studied is important and challenging. The proposed transformer-based architecture is a reasonable design for the application. The distinction between easy and hard prediction cases, which is usually overlooked, seems to be a novel idea that is effective for the problem. One strength of this work is to release an important dataset that will likely arouse more interest in the application. Also, thorough experimental evaluation involving many baselines and out-of-domain generalization study is another strength of this work. Nevertheless, quite a few questions were raised by the reviewers. We appreciate the effort of the authors in answering the questions, helping to clarify many points. Further experiments were also conducted during the rebuttal phase to help answer some of the questions.

The authors are highly recommended to take the comments and suggestions of the reviewers into consideration when revising their paper.